# Templated folding of the RTX domain of the bacterial toxin adenylate cyclase revealed by single molecule force spectroscopy

Han Wang[1,2], Guojun Chen[1] & Hongbin Li [1✉]

The RTX (repeats-in-toxin) domain of the bacterial toxin adenylate cyclase (CyaA) contains five RTX blocks (RTX-i to RTX-v) and its folding is essential for CyaA's functions. It was shown that the C-terminal capping structure of RTX-v is critical for the whole RTX to fold. However, it is unknown how the folding signal transmits within the RTX domain. Here we use optical tweezers to investigate the interplay between the folding of RTX-iv and RTX-v. Our results show that RTX-iv alone is disordered, but folds into a $Ca^{2+}$-loaded-β-roll structure in the presence of a folded RTX-v. Folding trajectories of RTX-iv-v reveal that the folding of RTX-iv is strictly conditional upon the folding of RTX-v, suggesting that the folding of RTX-iv is templated by RTX-v. This templating effect allows RTX-iv to fold rapidly, and provides significant mutual stabilization. Our study reveals a possible mechanism for transmitting the folding signal within the RTX domain.

[1] Department of Chemistry, University of British Columbia, Vancouver, BC V6T 1Z1, Canada. [2]Present address: State Key Laboratory of Precision Measuring Technology and Instruments, School of Precision Instrument and Optoelectronics Engineering, Tianjin University, 300072 Tianjin, P. R. China. ✉email: Hongbin@chem.ubc.ca

The bacterial toxin adenylate cyclase (CyaA) produced by the Gram-negative bacteria *B. pertussis* is the key virulence factor of whooping cough[1–5]. CyaA belongs to a large, so-called repeats-in-toxin (RTX) family of toxins, which include cytolytic toxins, metalloproteinases, and lipases[6–9]. RTX toxins all share a common feature—a C-terminal RTX domain that is composed of tandem nonapeptide repeats with a consensus sequence GGXGXDXXX (where X represents any amino acid residue) that are rich in glycine and aspartate and can bind $Ca^{2+}$. The RTX domains are disordered in the $Ca^{2+}$-depleted bacterial cytosol, but fold into a $Ca^{2+}$-loaded functional β-roll structure in the extracellular milieu[10–14], where the concentration of free $Ca^{2+}$ is in the mM range.

CyaA is synthesized in the bacterial cytosol as an inactive protein. After being secreted to the extracellular space by the type I secretion system (T1SS), CyaA binds $Ca^{2+}$ and folds into a $Ca^{2+}$-loaded functional toxin. CyaA is a bi-functional toxin and exhibits both hemolytic and cytolytic functions[15]. CyaA plays important roles in the early stages of respiratory tract colonization by *B. pertussis*[16]. After entering host cells, CyaA is activated by the binding of cytosolic calmodulin and converts cytosolic ATP into the key cellular signaling molecule cAMP in an unregulated fashion[12,17], leading to the complete disruption of cellular signaling and abolishment of the bactericidal functions of host phagocytes[13,16,18,19].

The highly efficient secretion and subsequent folding of CyaA are critical for CyaA toxin activity and the virulence of *B. pertussis*[11]. The secretion of CyaA starts from its C-terminus and the RTX domain of CyaA is the first region to be secreted after the secretion signal sequence. The RTX domain of CyaA contains 40 nonapeptide repeats that are arranged into five RTX blocks (RTX-i to RTX-v). It is now well-established that the $Ca^{2+}$-driven folding of the RTX domain plays important roles in accelerating the secretion of CyaA and mediating the binding of CyaA to host phagocytes by recognizing the complement receptor 3 (CR3, also known as $\alpha_M\beta_2$ integrin CD11b/CD18)[14,20–23]. Revealing the molecular features underlying the efficient secretion and folding of RTX domain is not only critical for a better understanding of the working mechanism of CyaA toxin but may also help develop new approaches to design therapeutics to combat *B. pertussis* and other major pathogens.

Recent studies showed that RTX-v folds vectorially in the direction from its C-terminus to N-terminus[10,11,24,25], coinciding with RTX-v's secretion direction. This inherent feature of RTX-v suggests that the folding of RTX-v is a co-secretional folding process that helps RTX-v be secreted efficiently. More importantly, it was discovered that disrupting the folding of the capping structure of RTX-v can disrupt the folding of the whole RTX domain (RTX-i-v) and thus ablate the CR3 binding site that is located between RTX-ii and RTX-iii[11], leading to the abolishment of CyaA toxin activity. Evidently, the capping structure of RTX-v has an allosteric regulation effect on the folding of RTX-v as well as RTX-i-v, including the CR3 binding site. However, the molecular mechanism underlying this allosteric regulation remains unknown. It was proposed that the capping structure serves as a scaffold for the RTX repeats in RTX-v to assemble into the β-roll structure[10,11,26]. This capping structure has been used to scaffold the folding of RTX-i-iii[27], and mutants of RTX-v with rearranged RTX repeats were also engineered[24]. In addition, it was found that a deletion variant of RTX-i-v, in which 267 residues (residue 1295–1561) encompassing the C-terminal part of RTX-iii, RTX-iv and N-terminal part of RTX-v were deleted, was folded with a properly constituted CR3 binding site[28]. Although these artificial constructs revealed insights into the role of the stacking of the RTX nonapeptide repeats in the assembly of the β-roll structure, it remains unknown how the folding signal is transmitted

between RTX blocks from the capping structure all the way to RTX-i within the RTX domain.

One of the main challenges in probing the transmitting mechanism of the folding signal in the RTX domain lies in the difficulties in distinguishing the folding of each individual RTX block that are tandemly arranged in the RTX domain by using traditional biophysical methods. Owing to its ability to monitor the folding–unfolding of each individual domain in a tandem modular protein at the single-molecule level, optical trapping (OT) technique may offer a unique approach to tackle this challenge. Due to its superb resolution in force and length, OT has evolved into a powerful tool to investigate the folding and unfolding mechanisms of proteins at the single-molecule level[29–33]. Using OT, we have investigated the folding mechanism of RTX-v in detail, and revealed key molecular features that help ensure the highly efficient secretion of RTX-v[25]. These studies laid a solid foundation for further investigations of the folding of the RTX domain. As the first step towards elucidating the molecular mechanism via which the folding signal transmits along the RTX domain, here we used OT to directly investigate the folding/unfolding mechanism of RTX-iv and its influence by the folding of its C-terminal neighboring RTX-v in the tandem repeats RTX-iv-v.

## Results

RTX-iv is 152 amino acid residues long (residues 1377–1528) and consists of ten nonapeptide repeats. In its $Ca^{2+}$-loaded crystal structure of RTX-iv-v, RTX-iv folds into a β-roll structure, which consists of 12 β-strands and 7 bound $Ca^{2+}$ ions (Fig. 1a, colored in red). The three nonapeptide repeats at the N-terminus of RTX-iv were not resolved in the 3D structure of RTX-iv-v and are presumably unfolded. $Ca^{2+}$ ions are bound within the two neighboring turns by charge-dipole interactions. The β strands are arranged in a tandem fashion and packed tightly onto the N-terminus of RTX-v (Fig. 1a). RTX-iv and RTX-v form a seemingly large contiguous β-roll structure[26]. To examine the transmitting of the folding signal between RTX-iv and RTX-v, we first examined the folding of RTX-iv alone.

**RTX-iv does not fold alone but can fold in the presence of folded RTX-v.** As shown in Fig. 2a, the far UV circular dichroism (CD) spectra of RTX-iv alone were similar in $Ca^{2+}$-free buffer and buffer containing 10 mM $Ca^{2+}$, and both were characterized by a negative band near 200 nm, indicating that RTX-iv alone is intrinsically disordered even in the presence of a super-physiological concentration of $Ca^{2+}$ (10 mM). To examine if RTX-iv can fold into any marginally stable folded structure, we used OT to examine its mechanical unfolding and folding. For OT experiments, we constructed a protein chimera NuG2-RTX-iv-NuG2, in which the well-characterized NuG2 was used as a fingerprint to facilitate the identification of single-molecule stretching event. The mechanical unfolding of NuG2 is characterized by a contour length increment ($\Delta Lc$) of ~18 nm, and an unfolding force of 30–50 pN at a pulling speed of 50 nm/s[30,34]. In a given force–distance (F–D) curve, the observation of two NuG2 unfolding events indicates a successful single-molecule stretching event of the protein chimera, and ensures that the F–D curve contains the mechanical signatures of RTX-iv. In the absence of $Ca^{2+}$ and presence of 10 mM $Ca^{2+}$, the F–D curves of NuG2-RTX-iv-NuG2 only displayed a pair of unfolding–refolding events of NuG2, and no mechanical (un)folding event of RTX-iv was observed (Fig. 2b), suggesting that RTX-iv itself behaved as an unfolded polypeptide chain. These results indicated that RTX-iv itself is intrinsically disordered, and the binding of $Ca^{2+}$ is not

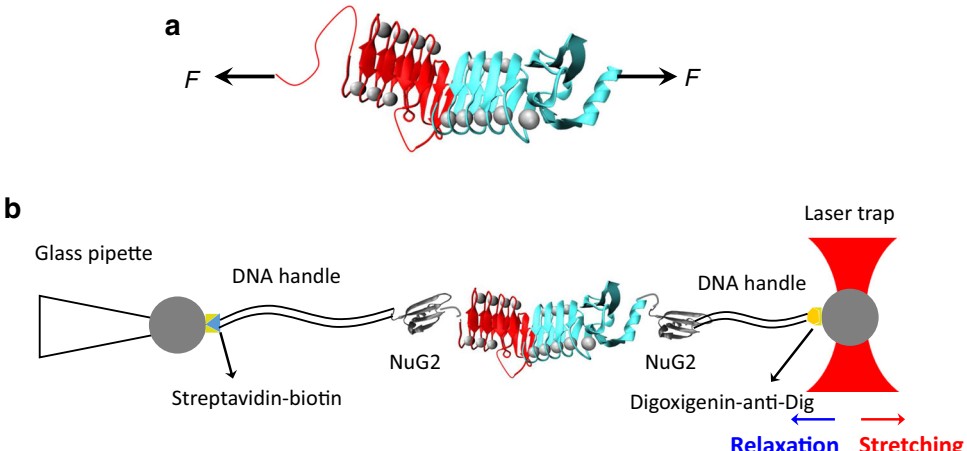

**Fig. 1 Probing the unfolding and folding of RTX-iv using OT. a** The 3D structure of RTX-iv-v (PDB: 6SUS). RTX-iv is colored in red and RTX-v is colored in cyan. $Ca^{2+}$ ions are shown as gray spheres. RTX-iv and RTX-v fold into a seemingly continuous β-roll structure. **b** Schematics of the OT experiment on NuG2-RTX-iv-v-NuG2. The protein chimera NuG2-RTX-iv-v-NuG2 is coupled with two DNA handles via thiol-maleimide chemistry. The DNA–protein complex is then attached to two polystyrene beads, which are functionalized with streptavidin and anti-digoxigenin, via specific ligand-receptor interactions. One bead is held by a glass micropipette and the other is trapped in the laser trap. By moving the position of laser trap, the target protein can be mechanically stretched or relaxed, and the force–distance relationship of the protein-DNA chimera can be measured.

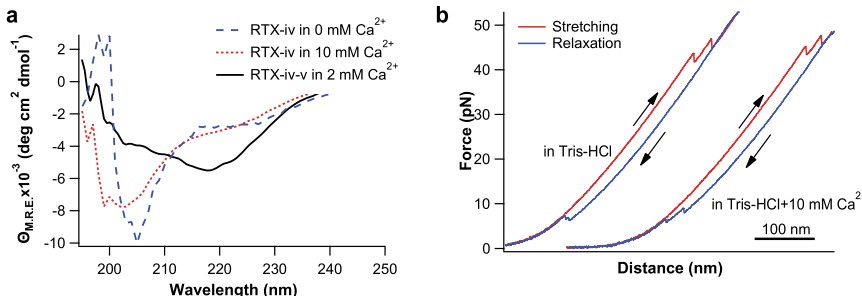

**Fig. 2 RTX-iv alone is intrinsically disordered. a** Far UV CD spectrum of RTX-iv in the absence of $Ca^{2+}$ and the presence of 10 mM $Ca^{2+}$. **b** Representative *F–D* curves of NuG2-RTX-iv-NuG2 in Tris-HCl (left) and Tris-HCl+10 mM $Ca^{2+}$ (right) at a pulling speed of 50 nm/s. Only the unfolding and refolding events of NuG2 domains were observed, suggesting that RTX-iv alone behaved as a disordered polypeptide.

enough for RTX-iv to fold, implying that other factors are required to help RTX-iv fold into its folded β-roll structure.

To test if RTX-v can help RTX-iv fold in solution, we engineered RTX-iv-v and used CD spectroscopy to examine its folding. As shown in Fig. 2a, the far UV CD spectrum of RTX-iv-v showed a strong band at ~217 nm, which is characteristic of the β-roll structure[11]. This result suggested that RTX-iv can fold into its $Ca^{2+}$-loaded folded structure in 2 mM $Ca^{2+}$ in the presence of RTX-v, a finding that is consistent with the crystallographic result on RTX-iv-v[26]. However, in the 3D structure of RTX-iv-v, RTX-iv and RTX-v form a large and seemingly contiguous β-roll structure[26,28], raising the question if RTX-iv and RTX-v are two "independent" domains or just one large domain.

**RTX-iv and RTX-v are two folded thermodynamic entities**. To examine if RTX-iv and RTX-v are independent domains, we engineered NuG2-RTX-iv-v-NuG2 and used single-molecule OT to examine the folding–unfolding mechanism of RTX-iv-v. Stretching NuG2-RTX-iv-v-NuG2 in the presence of 10 mM $Ca^{2+}$ resulted in *F–D* curves displaying four distinct unfolding events (Fig. 3a), in which the two unfolding events occurring at forces higher than 30 pN were due to the unfolding of the two NuG2 domains, and the two additional unfolding events at ~10 pN can be readily attributed to the unfolding of holo-RTX-iv-v.

Upon relaxation, four refolding events were clearly observed, with two occurring at ~8 pN, which were resulted from the folding of NuG2 domains, and the other two occurring at ~4 pN, which can be attributed to the folding of RTX-iv-v (Fig. 3a). Since the unfolding of RTX-iv-v occurred at significantly lower forces than NuG2, subsequent stretching–relaxation cycles were limited to below 20 pN so that only RTX-iv-v unfolded and folded, while NuG2 domains remained folded throughout (Fig. 3b).

From the *F–D* curves, it is evident that the unfolding and folding of RTX-iv-v showed two major events (colored in red and cyan, respectively), suggesting that RTX-v and RTX-iv may unfold and fold "independently". Fitting the force versus length change relationship for the two (un)folding events to the WLC model of polymer elasticity[35] measured a $\Delta Lc$ of $46.2 \pm 0.9$ nm (the data are presented as average ± standard deviation throughout this work) (cyan event) and $39.9 \pm 0.8$ nm (red event) (Fig. 3b, d), respectively. The $\Delta Lc$ of 46.2 nm is identical to the (un)folding signature of RTX-v, hence the (un)folding events colored in cyan were derived from the (un)folding of RTX-v. Accordingly, the (un)folding events colored in red can be assigned to the (un)folding of RTX-iv. The unfolding of RTX-iv occurred at ~10 pN and refolded at ~4 pN at a pulling speed of 50 nm/s (Fig. 3e). In comparison, RTX-v unfolded at ~13 pN and refolded at ~4 pN (Supplementary Fig. 1), and its unfolding and folding showed both two-state and three-state behaviors. These features are the

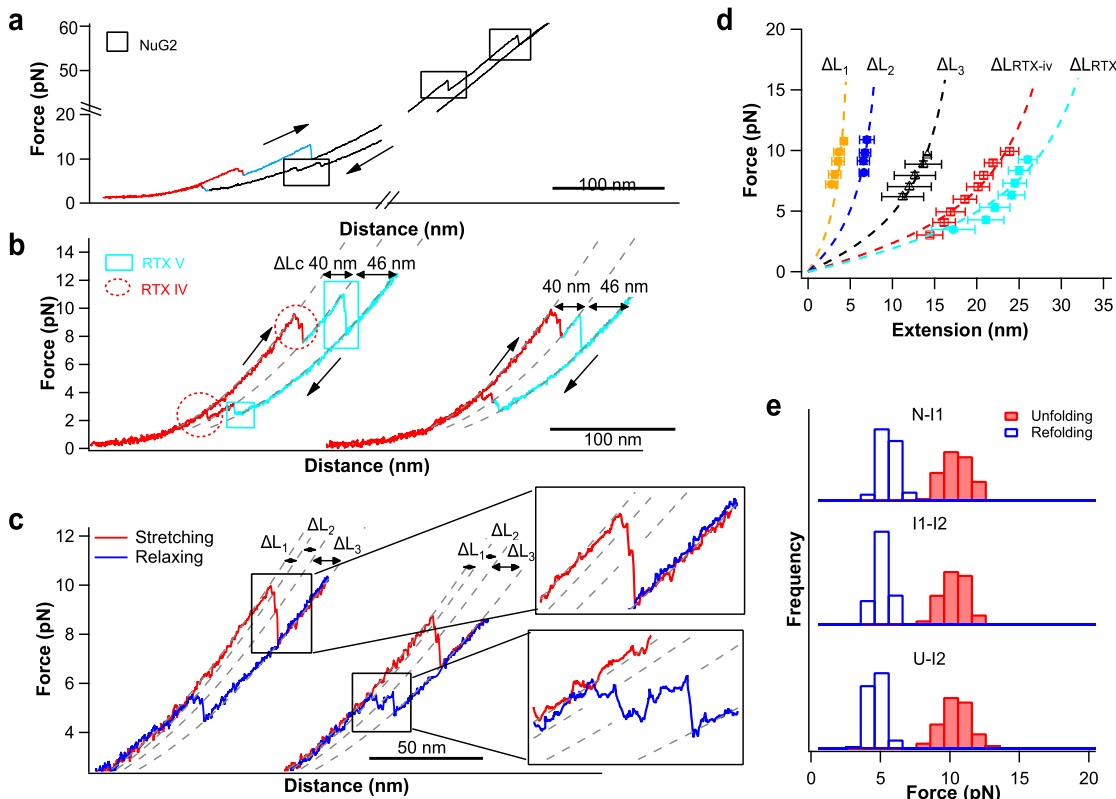

**Fig. 3 Holo-RTX-iv-v is mechanically stable and can fold rapidly. a** Representative *F–D* curves of the full-length NuG2-RTX-iv-v-NuG2. The unfolding and refolding events of NuG2 domains are colored in black and indicated by the squares, and the unfolding–folding of RTX-iv and RTX-v are colored in red and cyan, respectively. **b** Representative stretching and relaxation *F–D* curves showing the unfolding–folding of RTX-iv and RTX-v. The unfolding and folding events of RTX-iv are colored in red and indicated by circles, and the unfolding–folding events of RTX-v are colored in cyan and indicated by squares. For clarity, the two stretching–relaxation cycles are offset relative to each other. **c** *F–D* curves show that unfolding and folding of RTX-iv occur via a four-state pathway involving two intermediate states. The stretching–relaxation curves shown herein were limited to below 11–12 pN so that only RTX-iv unfolds and refolds, while RTX-v remained folded throughout. Insets show the zoomed view of the *F–D* curves showing the four-state unfolding and folding. Gray dashed lines in (**b**, **c**) are pseudo-worm-like chain (WLC) fits to the *F–D* curves. **d** Force–extension relationships of the (un)folding of RTX-iv-v. WLC fits (dashed lines) to the experimental data measured a $\Delta Lc$-RTX-iv of 39.9 ± 0.8 nm ($n = 37$) and $\Delta Lc$-RTX-v of 46.2 ± 0.9 nm ($n = 42$), respectively. WLC fits to the unfolding events involving intermediate states revealed $\Delta Lc1$ of 6.5 ± 0.9 nm ($n = 87$, from the native state to the first intermediate state, N-I1), $\Delta Lc2$ of 11.3 ± 0.7 nm ($n = 91$, from the first intermediate state to the second intermediate state, I1-I2), $\Delta Lc3$ of 23.5 ± 0.5 nm ($n = 115$, from the second intermediate state to the unfolded state, I2-U), respectively. The error bars indicate standard deviations. **e** Histograms of unfolding and refolding forces of RTX-iv at a pulling speed of 50 nm/s. The number of unfolding and refolding events is shown in Supplementary Table 1.

same as those of standalone RTX-v reported previously in ref. [25]. These results clearly indicated that in the presence of RTX-v, RTX-iv folded into a mechanically stable structure, and RTX-iv and RTX-v can be largely considered as two domains.

It is of note that WLC model cannot be used to fit *F–D* curves (see "Methods" for details) to obtain accurate measurements of $\Delta Lc$ for unfolding and folding events. However, WLC fits to *F–D* curves can serve as a rough guide allowing us to distinguish (un)folding events with different $\Delta Lc$ (Fig. 3b). We termed such fits pseudo WLC fits. It is evident that the unfolding events of RTX-iv and v can be readily distinguished by using such pseudo WLC fits.

To confirm this assignment, we compared the experimentally determined $\Delta Lc$ of 39.9 nm with that expected based on the 3D structure of RTX-iv in RTX-iv-v. RTX-iv (residues 1377–1528) consists of ten nonapeptide repeats. In the 3D structure of RTX-iv, only the seven nonapeptide repeats (residues 1410–1528) at the C-terminus fold into a β-roll structure, while the three nonapeptide repeats (residues 1377–1409) at the N-terminus are not resolved in the crystal structure of RTX-iv (pdb code: 6SUS)[26]. Unfolding of the β-roll structure of RTX-iv (residues 1410–1528) would result in a $\Delta Lc$ of 40.4 nm (119 aa × 0.36 nm/

aa – 2.4 nm = 40.4 nm, where 2.4 nm is the distance between I1410 and V1528), in good agreement with the experimentally measured $\Delta Lc$. This result indicated that the N-terminal three nonapeptide repeats are indeed unfolded.

To further corroborate these findings, we engineered a variant RTX-iv-T1411C-v, in which residue Thr1411, which is located at the N-terminal end of the β-roll structure of RTX-iv, was mutated to cysteine. By attaching the dsDNA handles to Cys1411 and the C-terminal Cys, we can stretch the segment encompassing residues 1411–1528 of RTX-iv together with the whole RTX-v. As shown in Fig. 4, stretching RTX-iv-v between Cys1411 and the C-terminus resulted in *F–D* curves that are the same as those of wild-type (wt) RTX-iv-v, in which the (un)folding of RTX-iv resulted in $\Delta Lc$ of ~40 nm. These results corroborated that the observed (un)folding events of RTX-iv are derived from the folded β-roll structure and that the three N-terminal nonapeptide repeats behave as an unfolded polypeptide chain.

**The folding and unfolding of RTX-iv involve two intermediate states.** A closer inspection of the (un)folding events of RTX-iv (Figs. 3c and 4b) revealed that the great majority (96%) of RTX-iv unfolded following a four-state pathway involving two short-

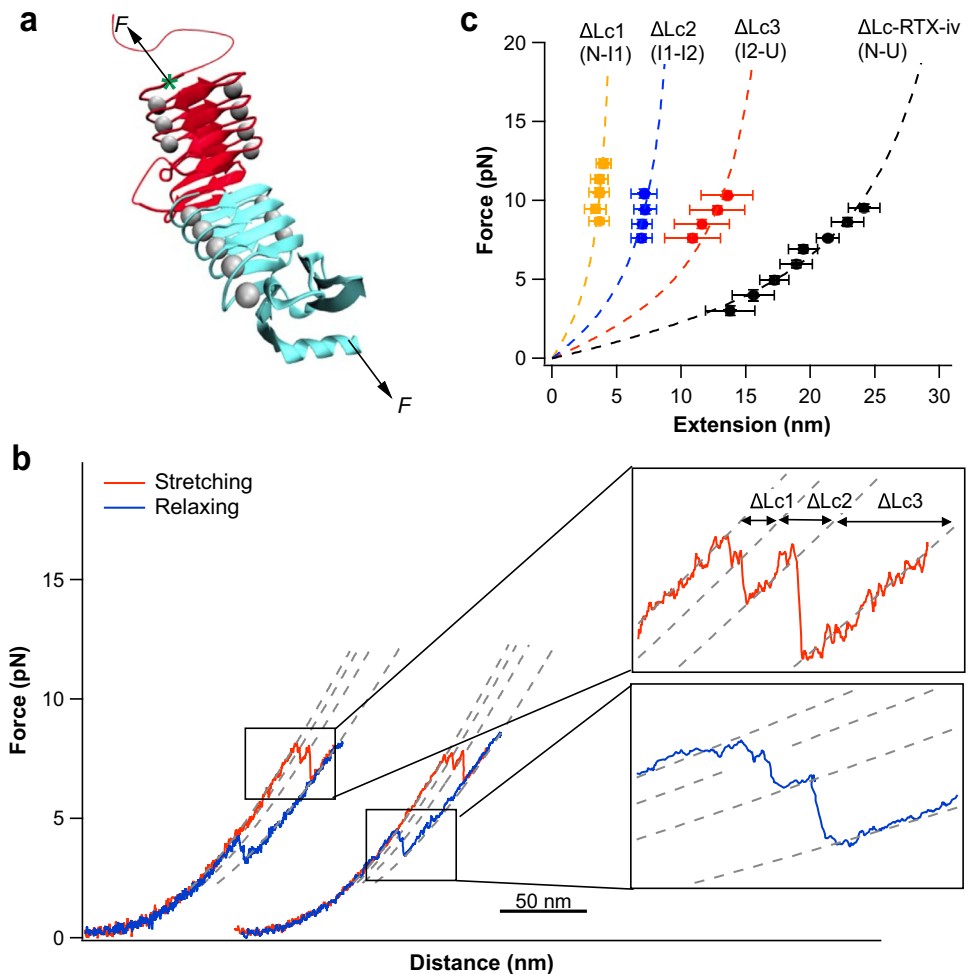

**Fig. 4 Mechanical unfolding/folding of RTX-iv-T1411C-v. a** Schematics of stretching RTX-iv-T1411C-v from the C-terminus and T1411C (indicated by the green asterisk). **b** Representative *F–D* curves of RTX-iv-T1411C-v, in which the molecule was stretched to unfold–fold RTX-iv-T1411C only. The unfolding and folding patterns are the same as those of wt RTX-iv. **c** Force–extension relationships of the unfolding of RTX-iv-T1411C. The WLC fits yielded a $\Delta Lc1$ ($n = 57$, N-I1) of $6.1 \pm 0.4$ nm, $\Delta Lc2$ ($n = 52$, I1–I2) of $11.0 \pm 0.5$ nm, $\Delta Lc3$ ($n = 64$, I2-U) of $23.1 \pm 0.6$ nm and $\Delta Lc$ ($n = 30$, RTX-iv-T1411C) of $39.7 \pm 0.8$ nm. The error bars indicate standard deviations.

lived intermediate states (I1 and I2) (Fig. 3c). WLC fits to the force–extension relationships yielded $\Delta Lc1$ of $6.5 \pm 0.9$ nm, $\Delta Lc2$ of $11.3 \pm 0.7$ nm, and $\Delta Lc3$ of $23.5 \pm 0.5$ nm, respectively (N→I1, I1→I2, and I2→U, Fig. 3c, d). And a small percentage of RTX-iv (~4%) was observed to visit only one intermediate state (either I1 or I2) en route to its folded state (Table 1 and Supplementary Fig. 2).

The folding of RTX-iv also occurred involving folding intermediate states. A large portion of RTX-iv folded involving two intermediate states (70.6%), while a small portion foldeds involving only one intermediate state (23.4%) or in an apparent two-state fashion (6%) (Table 1). The $\Delta Lc$ observed in the four-state folding pathway are the same as those in the four-state unfolding process, suggesting that unfolding and folding intermediate states are likely the same, and the unfolding and folding occur reversibly.

Although the observation of (un)folding trajectories involving different number of intermediate states could suggest the existence of multiple parallel (un)folding pathways, it is plausible that there exists only one unique (un)folding pathway for RTX-iv, i.e. the four-state pathway, and the trajectories with missing intermediate states were a result of the limited temporal resolution of our OT setup. Future OT experiments with improved temporal resolution will be needed to resolve this issue.

**Table 1 Numbers and percentage of each individual unfolding and folding pathway of RTX-iv.**

|  | N-I1-I2-U | N-I1-U | N-I2-U | N-U |
|---|---|---|---|---|
| Unfolding | 502 (96.4%) | 4 (0.8%) | 15 (2.9%) | 0 |
| Folding | 368 (70.6%) | 21 (4.0%) | 101 (19.4%) | 31 (6.0%) |

Taken together, these results indicated that RTX-iv folded into a β-roll structure in the presence of folded RTX-v and 10 mM $Ca^{2+}$, and RTX-iv and RTX-v behaved as two seemingly independent domains during their folding and unfolding. To fully characterize the biophysical properties of RTX-iv, future experiments will be needed to carry out experiments at varying $Ca^{2+}$ concentrations and/or ionic strength.

**The folding of RTX-iv is templated by the folding of RTX-v: direct evidence from folding trajectories of RTX-iv-v.** It is clear that RTX-iv can fold in the presence of folded RTX-v, suggesting that RTX-v may serve as a scaffold or template to facilitate the folding of RTX-iv, in other words, the folding of RTX-iv may be templated by the folding of RTX-v. However, little is known

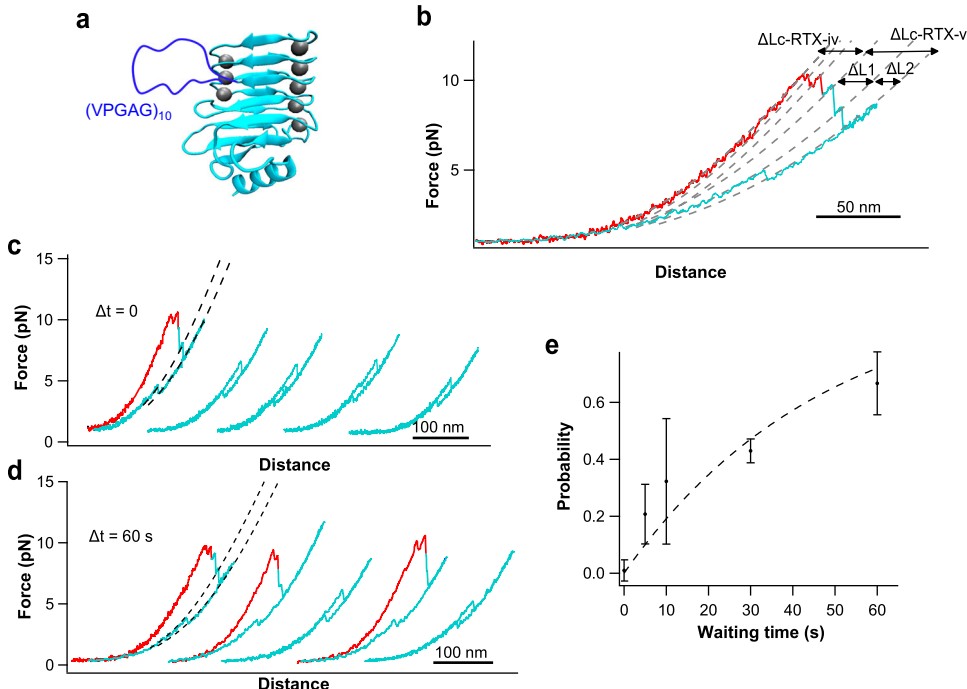

**Fig. 5 The folding of RTX-iv is conditional upon the folding of the whole RTX-v. a** Schematics of the structure of RTX-v$_{50}$. RTX-v$_{50}$ is constructed by inserting an intrinsically disordered elastin-like sequence (VPGAG)$_{10}$ between residue Leu1572 and Ser1573 in wt RTX-v. **b** A representative stretching–relaxation F–D curve of RTX-iv-v$_{50}$. The unfolding event of RTX-iv is colored in red, and the unfolding and partial refolding events of RTX-v$_{50}$ are colored in cyan. **c, d** Consecutive F–D curves showing the unfolding and refolding of RTX-iv-v$_{50}$. The unfolding event of RTX-iv is colored in red, and unfolding and refolding events of RTX-v$_{50}$ are colored in cyan. **c** No waiting time ($\Delta t$) was allowed between consecutive stretching–relaxation cycles, while in (**d**) $\Delta t$ was 60 s. In the relaxation curves of RTX-iv-v$_{50}$, only the refolding of C-FI at ~4 pN was observed, while no trajectory in which both RTX-iv and the N-terminal part of RTX-v refolded was observed. **e** The relationship of folding probability versus $\Delta t$ at zero force of the N-terminal part of RTX-v$_{50}$. Fitting the data to the first-order kinetics led to a folding rate constant of $(2.1 \pm 0.4) \times 10^{-2}$ s$^{-1}$ of the N-terminal part of RTX-v$_{50}$ at zero force. The number of events at $\Delta t = 0, 5, 10, 30, 60$ s are 172, 47, 35, 57, and 24, respectively. The error bars indicate standard deviations.

about this proposed templating effect. To test the validity of this possible templated folding effect, we used OT to examine the folding hierarchy in RTX-iv-v. If the folding of RTX-iv is indeed templated by the folding of RTX-v, the folding of RTX-iv-v should follow a strict hierarchical order with RTX-v folding always prior to RTX-iv.

As clearly indicated by the pseudo WLC fits in Fig. 3b, upon relaxation of the unfolded RTX-iv-v, RTX-v was observed to refold always prior to RTX-iv in 521 refolding trajectories without any exception, despite that the refolding of RTX-iv and RTX-v occurred at similar forces. We further verified that RTX-v refolded first during relaxation in consecutive stretching–relaxation curves (Supplementary Fig. 3). After RTX-iv-v had unfolded in the stretching curve (curve 1, Supplementary Fig. 3a), we relaxed the unfolded molecule until the first folding event occurred (curve 2, Supplementary Fig. 3a). Then the molecule was stretched again to unfold the refolded RTX block. From this subsequent unfolding event (which occurs at ~10 pN), we can accurately determine its $\Delta Lc$ and make the assignment. In addition, it is also straightforward to make the assignment by comparing curves 2 and 3 with curve 1. It is clear that the observed unfolding event in curve 3 matched the RTX-v unfolding event in curve 1 (Supplementary Fig. 3b). Therefore, the first refolding event can be assigned to the refolding of RTX-v without any ambiguity. These results confirmed that there exists a strict folding hierarchy, i.e. the folding of RTX-v always precedes the folding of RTX-iv, and this hierarchy is consistent with our proposed templating effect.

However, the observed folding hierarchy cannot prove the templating effect a priori, as the folding hierarchy is a necessary but not sufficient condition for the templating effect. To further

corroborate the existence of the templated folding of RTX-iv by RTX-v, we endeavored to prove that the folding of RTX-iv is conditional upon the folding of RTX-v. If the templating effect exists, slowing down the folding of RTX-v should also decelerate the folding of RTX-iv. Otherwise, the folding of RTX-iv should not be affected by a slow folding RTX-v.

To test if the folding of RTX-iv is conditional upon the folding of RTX-v, we engineered a loop-insertion variant of RTX-v to slow down the folding of RTX-v. In our previous work[25], we showed that RTX-v can fold and unfold via a C-terminal folding intermediate state (termed C-FI). We reason that inserting an unstructured loop outside the C-FI should slow down the folding of the whole RTX-v, due to the entropic penalty associated with closing an unstructured loop[36–39], but not affect the folding of the C-FI. For this purpose, we inserted a 50 residue long flexible loop, which is an intrinsically disordered elastin-like sequence (VPGAG)$_{10}$, in the loop connecting two β-strands (residue Leu1572 and Ser1573) in the N-terminal part of RTX-v, and the resultant loop-insertion variant of RTX-v is termed RTX-v$_{50}$ (Fig. 5a).

Far UV CD spectra of RTX-v$_{50}$ and RTX-iv-v$_{50}$ (Supplementary Fig. 4) showed that both constructs were folded and displayed the characteristic band of the β-roll structure. In addition, both CD spectra showed an increased content of random coils over wt RTX-v and RTX-iv-v, consistent with the incorporation of the disordered loop (VPGAG)$_{10}$.

Stretching RTX-iv-v$_{50}$ resulted in F–D curves with two groups of unfolding events (Fig. 5b): the first group (colored in red), showing a total $\Delta Lc$ of 40 nm, corresponded to the unfolding of RTX-iv. The second group (colored in cyan) can thus be assigned

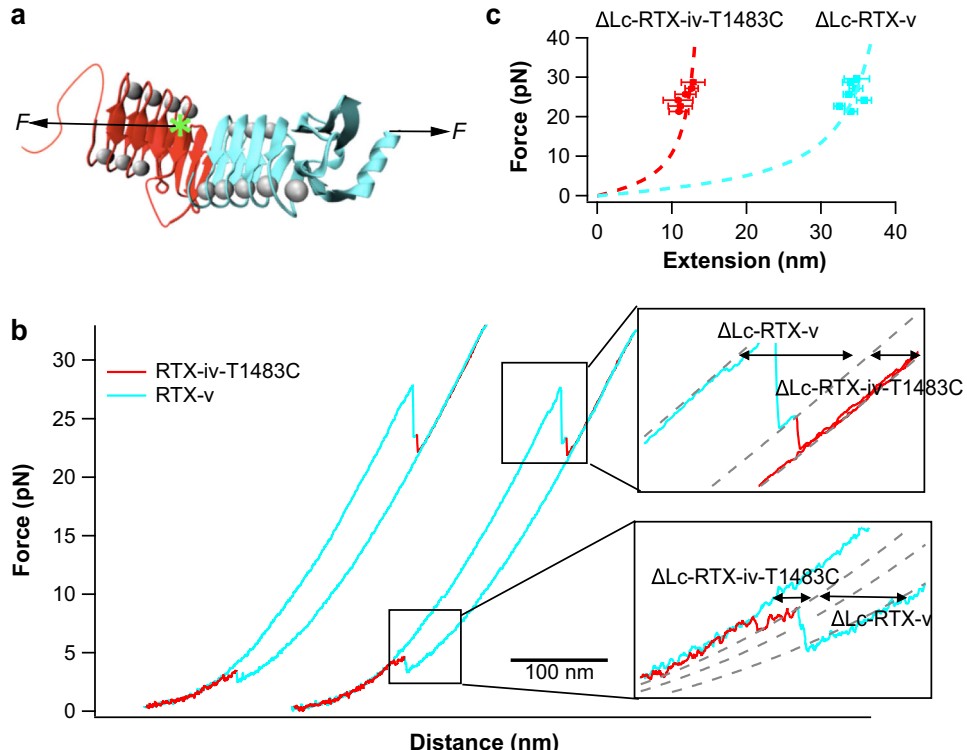

**Fig. 6 Mechanical unfolding/folding of RTX-iv-T1483C-v. a** Schematics of stretching RTX-iv-T1483C-v from its C-terminus and the residue 1483 (indicated by the green asterisk). **b** Representative force–distance curves showing the unfolding/folding of RTX-iv-T1483C-v at a pulling speed of 50 nm/s. The unfolding/folding of RTX-iv-T1483C is colored in red, and the unfolding/folding of RTX-v is colored in cyan. Both unfolding and folding initiate from RTX-v. **c** The WLC fitting of the unfolding of force–extension relationships of the unfolding of RTX-iv-T1483C (red) and RTX-v (cyan), respectively. The WLC fits yielded a $\Delta Lc$ of 16.1 ± 0.3 nm ($n = 105$) for RTX-iv-T1483C, and $\Delta Lc$ of 45.2 ± 0.7 nm ($n = 105$) for RTX-v, respectively. The error bars indicate standard deviations.

to the unfolding of RTX-v$_{50}$. The unfolding of most RTX-v$_{50}$ occurred in two steps and showed a total $\Delta Lc$ of 64 nm ($\Delta Lc1 = 36$ nm and $\Delta Lc2 = 28$ nm). The total $\Delta Lc$ of wt RTX-v is 46 nm, and the inserted flexible loop of 50 residues has a contour length of 18 nm. Hence, it is expected that unraveling RTX-v$_{50}$ should result in a $\Delta Lc$ of ~64 nm ($46+18 = 64$ nm), in close agreement with the measured total $\Delta Lc$ of RTX-v$_{50}$. As expected, the $\Delta L2$ of the C-FI (~28 nm) was not affected by the loop insertion, but the $\Delta L1$ for the N-terminal part (from native state to C-FI) increased from 18 nm in wt RTX-v to 36 nm in RTX-v$_{50}$.

The loop insertion has a pronounced effect on the folding kinetics of RTX-v (Fig. 5b–d). Upon relaxation of the unfolded RTX-iv-v$_{50}$, only the folding events of the C-FI were observed at ~4 pN, and no folding events of the N-terminal part of RTX-v$_{50}$ or RTX-iv were observed (Fig. 5b). As expected, the folding kinetics of C-FI in RTX-v$_{50}$ was similar to that in wt RTX-v, as the loop insertion is outside of C-FI and bears no influence on the folding of C-FI. In contrast, the loop insertion dramatically decelerated the folding of the N-terminal part of RTX-v$_{50}$, due to the entropic penalty in closing the inserted long loop during the folding of the N-terminal part. In wild-type RTX-v, the N-terminal part folds readily at ~4 pN during relaxation; but in RTX-v$_{50}$ did not refold during relaxation. Moreover, during the consecutive stretching and relaxation cycles (with a waiting time $\Delta t = 0$ between cycles), only the unfolding and refolding of C-FI were observed (Fig. 5c), suggesting that the N-terminal part of RTX-v$_{50}$ did not refold even at zero force. This result also strongly indicated that RTX-iv did not fold even in the presence of a folded C-FI.

To quantitatively measure the folding kinetics of RTX-v$_{50}$, we relaxed the protein to zero force and measured the folding probability of RTX-v$_{50}$ at zero force as a function of waiting time $\Delta t$. This approach is similar to the double pulse approach used in atomic force microscopy-based single force spectroscopy experiments for characterizing the folding kinetics of proteins[40,41]. For these experiments on RTX-iv-v$_{50}$, it is important to point out that upon relaxing to zero force, in the relaxed molecule, C-FI was folded but both of the N-terminal part of RTX-v and RTX-iv were unfolded. As shown in Fig. 5d, when $\Delta t = 60$ s, the folding probability of RTX-v$_{50}$ increased considerably. In about 60% of the trajectories, RTX-iv and RTX-v refolded during the 60-s waiting time, as evidenced by their unfolding events in the subsequent stretching curves. Fitting the folding probability vs $\Delta t$ (Fig. 5e) to a first-order rate law, we estimated the folding rate constant of the N-terminal part of RTX-v$_{50}$ to be 0.02 s$^{-1}$ at zero force, in sharp contrast with that in wt RTX-v ($2.8 \times 10^3$ s$^{-1}$)[25]. The unfolding event of RTX-iv, which indicated that RTX-iv had refolded at zero force during the waiting time $\Delta t$, only appeared with the unfolding event of the fully folded RTX-v$_{50}$, and we did not observe any trajectory in which RTX-v$_{50}$ had refolded but RTX-iv did not. Similarly, we did not observe any trajectory in which only CF-I and RTX-iv refolded. This result suggested that RTX-iv could refold rapidly right after RTX-v$_{50}$ had folded, and clearly indicated that the folding of RTX-iv is conditional upon the folding of the whole RTX-v.

Together with the strict folding hierarchy of RTX-iv-v, these results demonstrated that the folding of RTX-iv is templated by the folding of RTX-v. And this templating effect requires the

folding of the whole RTX-v, and the folding of C-FI alone is not sufficient to template the folding of RTX-iv.

Building upon these insights, we analyzed the unfolding and folding data of wt RTX-iv-v and determined the folding kinetics of RTX-iv in the presence of a fully folded RTX-v by measuring the folding rate constants of RTX-iv as a function of the stretching force. Fitting the experimental data (Supplementary Fig. 5) to the Bell-Evans model[42,43] allowed us to estimate the folding rate constant at zero force (Supplementary Table 1). It is evident that the folding of RTX-iv is fast in the presence of a fully folded RTX-v.

**Mutual stabilization between RTX-iv and RTX-v.** Since RTX-iv and RTX-v behave as two domains, RTX-iv-v can be understood from a perspective of protein–protein interactions. Evidently, RTX-iv is stabilized by RTX-v. However, different from typical protein–protein interactions, it is puzzling to observe that RTX-v does not appear to be stabilized by the folded RTX-iv, as the unfolding force of RTX-v in RTX-iv-RTX-v (~13 pN) is almost the same as that in the standalone RTX-v (~13 pN)[25]. The unfolding hierarchy of RTX-iv-v provided some valuable insights into this puzzling observation.

In 521 unfolding trajectories, RTX-iv-v was observed to always follow a strict hierarchical order (Fig. 3b), with RTX-iv unfolding first followed by RTX-v without any exception, although the unfolding forces of RTX-iv (~10 pN) and RTX-v (~13 pN) were comparable and their distributions had a significant overlap (Supplementary Fig. 6). It is well-established that the mechanical unfolding of tandem modular proteins is strictly dependent upon their mechanical stability, with the mechanically weakest domain unfolds first and the strongest domain unfolds last[44,45]. For proteins like RTX-iv-v that have similar unfolding forces, there should not exist a strict unfolding hierarchy between RTX-iv and RTX-v. A plausible explanation for this observed peculiar mechanical unfolding hierarchy is that RTX-v is stabilized by the folded RTX-iv to the extent that its unfolding force is much higher than that of RTX-iv, consequently the mechanically weaker RTX-iv always unfolds first. After RTX-iv has unfolded, the stabilization effect on RTX-v by the folded RTX-iv no longer exists and RTX-v unfolds at the same force as the standalone RTX-v, leading to the apparent lack of stabilization of RTX-v by RTX-iv and the peculiar unfolding hierarchy. To validate this possible mechanism, we intended to directly stretch RTX-v and the interface between RTX-iv and RTX-v. For this purpose, we engineered two Cys variants of RTX-iv, T1483C and F1465C.

The variant RTX-iv-T1483C-v allowed us to stretch the segment of RTX-iv from residue 1483 to 1528 as well as the whole RTX-v. In this construct, the N-terminal part of RTX-iv is not subject to the stretching force and provides stabilization to the whole RTX-iv (Fig. 6a). In this experiment, we used the construct RTX-iv-T1483C-v, which does not contain any NuG2 domain. Stretching RTX-iv-T1483C-v resulted in $F$–$D$ curves with two clear unfolding events (Fig. 6b). Fitting the force–extension relationship to the WLC model revealed that the first unfolding event (colored in cyan) showed a $\Delta Lc$ of 45 nm and resulted from the unfolding of RTX-v, and the second unfolding event (colored in red) showed a $\Delta Lc$ of 16.1 nm (Fig. 6c), which is in good agreement with the expected $\Delta Lc$ from stretching RTX-iv between residues 1483 and 1529. It is important to note that the unfolding of RTX-v occurred at ~30 pN at a pulling speed of 50 nm/s, significantly higher than that of RTX-v alone or in wt RTX-iv-v when stretched from its N and C-termini (~13 pN). In addition, RTX-v always unfolded prior to RTX-iv-T1483C. Likewise, OT experiments on the variant RTX-iv-F1465C-v, which does not contain any NuG2

domain, revealed similar results, i.e. RTX-v was observed to unfold at ~30 pN prior to the unfolding of RTX-iv-F1465C (Supplementary Fig. 7). These results clearly indicated that once folded, RTX-iv has a considerable stabilization effect on RTX-v: $\Delta\Delta G_{N-U}$ is estimated to be ~5.9 $k_B T$ (assuming the folding rate constant of RTX-v is the same, $\Delta\Delta G_{N-U} = ln\ (\alpha_0'/\alpha_0)$, where $\Delta\Delta G_{N-U}$ is the stabilization of the thermodynamic stability, $\alpha_0$ and $\alpha_0'$ are the intrinsic unfolding rate constant of the standalone RTX-v and RTX-v in RTX-iv-T1483C-v, respectively). In contrast, the thermodynamic stabilization of RTX-iv by RTX-v is estimated to be around 15 $k_B$T. Evidently, there exists a significant mutual stabilization effect between RTX-iv and RTX-v.

The results on these two variants also corroborated the templating effect of RTX-v on RTX-iv, as RTX-v always folded prior to RTX-iv, despite the unfolding hierarchy between RTX-iv and v was reversed. In addition, these two variants also provided an invaluable opportunity for us to determine the intrinsic stability of RTX-iv. In these two variants, RTX-v unfolded prior to RTX-iv. As soon as RTX-v unfolded, RTX-iv unfolded soon afterward. The lifetime of RTX-iv in both cases, which can be measured from the $F$–$D$ curves (Supplementary Fig. 8a), were very short (Supplementary Fig. 8b). From the force-dependence of unfolding rate constant, the intrinsic unfolding rate constant $\alpha_0$ of RTX-iv at zero force in the absence of folded RTX-v was estimated to be 0.11 s$^{-1}$ (Supplementary Fig. 8c), significantly larger than $\alpha_0$ of RTX-iv in the presence of folded RTX-v (1.6 × 10$^{-3}$ s$^{-1}$, Supplementary Table 1), highlighting the critical role of RTX-v in stabilizing the folded RTX-iv.

## Discussion

The secretion and subsequent folding of the RTX domain of CyaA play critical roles in the toxin activity of CyaA and the virulence of *B. pertussis*. The secretion starts from its C-terminus and the capping structure at the C-terminal end of RTX-v is essential for the folding of the whole RTX domain. Although it was shown the capping structure can scaffold the RTX repeats of RTX-v to fold into the β-roll structure[11,21,24,26,28], it is unknown how the folding signal is transmitted along the RTX domain from RTX-v to RTX-i. Our OT experiments provided the first insight into this important mechanism. Our results showed that the folding of RTX-iv is templated or scaffolded by the folded RTX-v, *i.e.* the folding of RTX-iv is fully conditional upon or gated by the folding of RTX-v. RTX-iv will not fold into its stable folded β-roll structure unless RTX-v has folded. However, in the presence of folded RTX-v, RTX-iv can fold rapidly with an apparent rate constant of ~10$^3$ s$^{-1}$ into its β-roll structure. It is important to note that this templating effect requires the fully folded RTX-v, suggesting that the interface between RTX-v and RTX-iv is critical for the folding and stabilization of RTX-iv. However, the specific residues at the interface that are responsible for this templating effect remain to be mapped, and future work will be needed to fully elucidate the molecular mechanism underlying this templating effect.

Our results also demonstrate the unique capability of OT to study the templated folding of RTX-iv-v. By using the contour length increment as a fingerprint, OT allowed us to identify and assign the unfolding and folding events to specific RTX block in an unambiguous fashion, and establish the folding and unfolding order of the two RTX blocks. Using this method, it should become feasible to investigate the folding mechanism of the full-length RTX domain (RTX-i-v), which represents a significant challenge to address using traditional biophysical techniques.

The templating effect observed in RTX-iv-v also has important implications for the secretion mechanism of the RTX domain. Upon secretion (in the direction of C- to N-terminus), the

intrinsically disordered RTX domain binds extracellular $Ca^{2+}$ and folds into its functional β-roll structure. The biophysical features of RTX-v, including its intrinsically disordered conformation in the $Ca^{2+}$-depleted bacterial cytosol, fast folding kinetics in the extracellular space, the ability to fold against a stretching force, and the mechanical stability of the folded state, ensure the efficient secretion of RTX-v[11,25]. The templated folding mechanism also ensures the efficient secretion of RTX-iv. On the one hand, the intrinsically disordered RTX-iv can be translocated by the T1SS machinery without the need to overcome any enthalpic resistance, and the folded RTX-v provides a template for RTX-iv to fold rapidly as soon as it emerges out of the T1SS conduit. The folded RTX-iv is mechanically stable and can thus prevent its backsliding. In addition, the ability of RTX-iv to fold against a residual force enables RTX-iv to generate a stretching force during its folding to further facilitate the translocation of the polypeptide chain in the T1SS channel. On the other hand, the folding of RTX-iv can in turn further stabilize RTX-v, serving as an additional reinforcement step for the secreted and folded RTX-v. The reinforcement eliminates any possibility that RTX-v unfolds under a stretching force to trigger a cascade unfolding of RTX-iv, and ensures the stability of the secreted and folded RTX-iv and RTX-v. These unique mechanical designs in the structure of RTX blocks help CyaA toxin achieve highly efficient secretion and folding of the RTX domain. Finding efficient means to disrupt these mechanical features will likely open up a new avenue to developing new therapeutics to combat *B. pertussis* and other pathogens.

## Methods

**Protein engineering.** The gene encoding RTX-iv with restriction sites (5′ *Bam*HI and 3′ *Bgl*II and *Kpn*I) was custom synthesized (GeneScript). RTX-iv gene was first subcloned into the pUC19 vector via *Bam*HI and *Kpn*I restriction sites. Then the gene encoding RTX-v with restriction sites (5′ *Bam*HI and 3′ *Bgl*II and *Kpn*I) was inserted into the construct pUC19/RTX-iv plasmid by alternating enzyme digestion and ligation steps to construct pUC19/RTX-iv-RS-v, where RS are arginine and serine resides generated by the reconstitution between the *Bgl*II and *Bam*HI restriction sites. The additional amino acid residues RS were then removed to obtain the construct pUC19/RTX-iv-v using megaprimer approach of site-directed mutagenesis. Following our well-established iterative molecular biology strategy[46], we constructed the gene of NuG2-RTX-iv-NuG2 and NuG2-RTX-iv-v-NuG2. The genes were then subcloned into a modified expressing vector pQE80L-CC[25,30], to build pQE80L/Cys-NuG2-RTX-iv-NuG2-Cys and pQE80L/Cys-NuG2-RTX-iv-v-NuG2-Cys, respectively.

The genes encoding three mutants of RTX-iv-v (T1411C, F1465C, and T1483C) were obtained by megaprimer approach of site-directed mutagenesis. These genes were then subcloned into a modified expressing vector pQE80L, which carried a cassette allowing to add one cysteine residue only at the C-terminus of the target protein, to build pQE80L/RTX-iv-v mutant-Cys. The gene encoding RTX-$v_{50}$ consisted of three components: N-terminal part of RTX-v (from residue 1529 to 1572), elastin-like sequence $(VPGAG)_{10}$ and C-terminal folding intermediate state (C-FI, from residue 1573 to 1681). These genes were obtained by polymerase chain reaction amplification method and ligated together to the target gene encoding RTX-$v_{50}$ following the iterative molecular biology strategy. All the constructed genes were confirmed by DNA sequencing.

The engineered proteins were overexpressed in *Escherichia coli* strain DH5α at 37 °C in 250 mL 2.5% LB media with 100 mg/L antibiotics ampicillin. 1 mM isopropyl-β-D-1-thiogalactopyranoside (IPTG, Thermo Fisher Scientific, Waltham, MA) was added to induce the protein overexpression at the optical density 0.7–0.8, and the protein overexpression continued for 4 hours at 37 °C. The bacteria cell pellets were harvested by centrifugation at 5,000 r.p.m. (4360×g) at 4 °C for 10 min, and resuspended in 10 mL phosphate-buffered saline (PBS, 10 mM, pH 7.4) buffer. In total, 10 µL protease inhibitor cocktail (SIGMA-ALDRICH, St. Louis, MO), 50 µL 100 mg/mL lysozyme from egg white (SIGMA-ALDRICH, St. Louis, MO), 1 mL 10% (w/v) Triton X-100 (VWR, Tualatin, OR), 50 µL 1 mg/mL DNase I (SIGMA-ALDRICH, St. Louis, MO) and 50 µL 1 mg/mL RNase A (Bio Basic Canada Inc, Markham, ON) were added for the cell lysis. The lysis reaction was kept at 4 °C for 40 min. The supernatant containing the target protein was then isolated by centrifugation at 10,000 r.p.m. (8,720×g) at 4 °C for 1 h, and the protein was purified by $Co^{2+}$ affinity column with TALON His-tag purification kit (TaKaRa Bio USA Inc, Mountain View, CA). The protein was eluted and stored in elution buffer (10 mM PBS, 300 mM NaCl, 250 mM imidazole). The purified apo-form RTX-iv-v polyprotein (free of $Ca^{2+}$) was at a concentration of ~1.0 mg/mL and stored at −20 °C.

**Circular dichroism spectroscopy experiments.** The circular dichroism (CD) spectra were measured by a Jasco Model J810 spectro-polarimeter flushed with nitrogen gas. A quartz cuvette with a path length of 0.1 cm was used to contain protein samples. The protein samples were dissolved in Tris-HCl buffer (20 mM, pH 7.4) and the same Tris-HCl buffer supplemented with 2 or 10 mM $CaCl_2$. The final concentration of the protein was ~10 µM. The spectra were recorded at a scan rate of 50 nm/min and were corrected for buffer contributions.

**Preparation of DNA–protein chimera.** DNA handles were prepared via the method described previously[25,30,32]. Two DNA handles were obtained via regular PCR amplification. The template was pGEMEX-1 plasmid and the modified primers were purchased from Integrated DNA Technologies (IDT Inc, San Jose, CA)[25]. After the PCR amplification, the DNA handles were purified by QIAquick PCR purification kit (QIAGEN, Germantown, MD) and allowed to react with 4-(N-Maleimidomethyl) cyclohexane carboxylic acid N-hydroxysuccinimide ester (SMCC, SIGMA-ALDRICH, St. Louis, MO) overnight to bear one maleimide group. The DNA handles were then allowed to couple with the freshly expressed proteins via thiol-maleimide interactions overnight to form the DNA–protein chimera. The DNA–protein chimera was diluted to ~10 nM and stored at −80 °C.

**Single-molecule OT experiments.** The optical tweezers experiments were carried out on the Minitweezers setup (http://tweezerslab.unipr.it/cgi-bin/home.pl)[47]. In total, 1 µL of streptavidin-coated polystyrene beads (1% w/v 1 µm, Spherotech Inc, Lake Forest, IL) was diluted by 3 mL Tris-HCl buffer (20 mM Tris, 150 mM NaCl, pH 7.4) and injected into a fluid chamber. The optical trap was used to capture one single streptavidin-coated bead and moved near the tip of a micropipette in the chamber. By applying a vacuum through the pipette, the bead could be fixed firmly. Then 1 µL of the diluted DNA–protein chimera sample was mixed with 4 µL of polystyrene beads coated with anti-digoxigenin (0.5% w/v, 2 µm, Spherotech Inc, Lake Forest, IL). The coupling reaction between the DNA–protein chimera and polystyrene beads was kept at room temperature for 30 min. Then the beads were also diluted by 3 mL Tris-HCl buffer and injected into the chamber. One single anti-digoxigenin bead was captured and brought near the fixed streptavidin bead to enable the biotin-streptavidin interaction between the DNA handles and the bead. Once the bead-DNA–protein dumbbell was established, by moving the position of the bead trapped in the laser beam, the target protein could be stretched (or relaxed) to trigger the mechanical unfolding (or folding) (Fig. 1b). For experiments on holo-RTX, Tris-HCl buffer (20 mM Tris, 150 mM NaCl, pH 7.4) supplemented with 10 mM $Ca^{2+}$ was used.

**Measuring force–extension relationships.** In single-molecule force spectroscopy studies, WLC model of polymer elasticity is widely used to fit the force–extension curves to measure the contour length change during protein unfolding and folding:

$$F = \frac{k_B T}{p}\left(\frac{1}{4(1-\frac{x}{L})^2} + \frac{x}{L} - \frac{1}{4}\right) \tag{1}$$

where $F$ is the force, $p$ is the persistence length, $x$ is the extension, $L$ is the contour length of the polymer chain, and $k_B T$ is the thermal energy.

Stretching the protein-DNA chimera resulted in force–distance ($F$–$D$) curves, in which the distance contains the contribution of the extension of the protein-DNA construct as well as the compliance of the optical trap. Thus, the $F$–$D$ curves cannot be directly fitted to the WLC model of polymer elasticity to measure the $\Delta Lc$ of protein unfolding. $F$–$D$ curves can be converted to force–extension ($F$–$E$) curves if the OT stiffness is known[47,48]. Due to the large size of the beads (2 µm) and the short dsDNA handles used in our OT measurements, the OT stiffness of the MiniTweezers may vary at different forces (http://tweezerslab.unipr.it/cgi-bin/documents.pl/Show?_id= 5eb0&sort=DEFAULT&search=&hits). In order to convert $F$–$D$ curves to $F$–$E$ curves, it becomes necessary to measure the stiffness of the optical trap in the MiniTweezers at different forces with each tethered molecule (http://tweezerslab. unipr.it/cgi-bin/documents.pl/Show?_id=5eb0&sort=DEFAULT&search=&hits), adding practical difficulties in the experiments. Instead of converting $F$–$D$ to $F$–$E$ curves, we employed a different, well-established strategy to measure the $\Delta Lc$ of protein unfolding[32,34,49]. For a given unfolding event with a contour length increment of $\Delta Lc$ in the $F$–$D$ curve, the length increase at a given force $F$ gives the extension $x$ of the unfolded polypeptide chain with a contour length of $\Delta Lc$. By fitting the measured force–extension relationship to the WLC model, we can measure the persistence length and $\Delta Lc$ of the unfolded polypeptide chain being released from the unfolding.

Although fitting WLC model (Eq. (1)) to the $F$–$D$ curves cannot provide accurate information on $\Delta Lc$ and persistence length, these "fits" can be used for eye-guiding purposes in helping identify unfolding or refolding events, and provide a rough qualitative estimate of $\Delta Lc$, which can be used to help distinguish unfolding/folding events with different $\Delta Lc$.

**Extracting kinetics of unfolding/folding of proteins from *F–D* curves.** We used the method proposed by Oesterhelt et al.[50] to measure the force-dependent unfolding/folding rate constants of proteins from constant pulling velocity experiments. Briefly, the $F$–$D$ were divided into time windows ($\delta t$) that are small enough so that the force can be considered constant within the time window. The

probability of protein folding/unfolding within $\delta t$ can be calculated as $P(F) = N(F)/M(F)$, where $N(F)$ is the total number of all the folding or unfolding events at a force of $F$, and $M(F)$ is the total number of time windows at a force of $F$. The rate constant of protein folding/unfolding at $F$ can then be calculated as $k(F) = P(F)/\delta t$.

**Reporting summary**. Further information on research design is available in the Nature Research Reporting Summary linked to this article.

## Data availability

The data that support the findings of this study are available from the corresponding author upon reasonable request. The protein structure presented in Fig. 1a is available via the PDB 6SUS. Source data are provided with this paper.

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

## Acknowledgements

This work is supported by the Natural Sciences and Engineering Research Council of Canada Discovery Grant (RGPIN-2020-06024) and the Canadian Institutes of Health Research Project Grant to H.L.

## Author contributions

H.W. and H.L. designed the experiments. H.W. and G.C. performed the experiments. H.W. analyzed the data. H.W. and H.L. wrote the paper.

## Competing interests

The authors declare no competing interests.
