## [Peer Review File · Nature Communications]

REVIEWER COMMENTS

Reviewer #1 (Remarks to the Author):

This paper titled "Templated Folding of the RTX Domain of the Bacterial Toxin Adeylate Cyclase Revealed by Single Molecule Force Spectroscopy" is a follow-up work by the authors, who previously had used single-molecule optical tweezers studies to characterize the folding/unfolding and mechanical properties of the C-terminal repeats-in-toxin (RTX) block vdomain (residue 1529-1681) from the bacterial toxin adenylate cyclase (CyaA) (Wang et al, JACS 2019). Here in this new work, Wang and Li have extended their survey on the two tandem domains, i.e., block iv and v (residue 1377-1681), in the C-terminus of CyaA and systematically interrogated how these two domains modulate each another's folding/unfolding pathway as well

as structural/mechanical stability. Their experimental results indicate that: (i) the two domains, RTX-iv and RTX-v, fold in a strict hierarchical fashion, such that folding of RTX-v always precedes folding of RTX-iv; (ii) folding of RTX-iv is conditional upon the folding of RTX-v, such that introducing an unstructured loop to paralyze folding of the N-terminal part of RTX-v, in turn, prevents folding of RTX-iv; and (iii) folded RTX-iv imparts mechanical strength to RTX-v, such that stretching RTX-v and the C-terminus of RTX-iv while keeping the N-terminus of RTX-iv folded dramatically increases the force (13 to 30 pN) required to unfold RTX-v. Based on these results, the authors propose that the folding of RTX-iv is "templated" by its partner RTX-v, due to interfacial interactions between the two domains. Ultimately, the authors mention that these observations have implications for the development of therapeutics against the pathogenicity of CyaA, a key virulence factor of whooping cough. Overall, the work of this paper is of sound quality, the layout is clear and easy to comprehend, and the results are of great interest to the protein folding field, and with implications on how the folding of CyaA toxin can be specifically impaired as one potential therapeutic approach. We would recommend this paper for publication in Nature Communications after the authors address the following issues and complete the modifications/edits as suggested below:

Major Comments

1. The authors indicated, based on their WLC fits to unfolding and refolding traces of the RTX-iv-v construct, that they observe both, only one, or none of the two intermediates along the transition paths of RTX-iv. Based on this observation, they suggest that RTX-iv folds and unfolds via multiple parallel pathways: $N \leftrightarrow I1 \leftrightarrow I2 \leftrightarrow U$, $N \leftrightarrow I1 \leftrightarrow U$, $N \leftrightarrow I2 \leftrightarrow U$, or $N \leftrightarrow U$. An alternative explanation of this observation could be that there is only one unique reversible pathway for folding and unfolding of RTX-iv (i.e. $N \leftrightarrow I1 \leftrightarrow I2 \leftrightarrow U$) but, since the temporal resolution of the optical tweezers experiments is finite, the lifetime of one (or both) of the intermediate states, I1/I2, in certain trajectories may fall shorter than the detection limit (say ~ 10 ms) and therefore the observed transition path appears to have fewer intermediates in a small minority of cases. This scenario is - equally plausible as the multiple parallel pathways the authors proposed and - consistent with the observation that the respective ΔLc 's remain the same for I1 and I2 in the four-state as well as three-state dataset. The authors should include in the supplementary the representative F-D curves for these "three-state pathways" to assist this discussion. Unless the authors can further characterize these intermediate states along the (un)folding pathway of RTX-iv that differ significantly from I1, I2 observed in the most common 4-state pathway (namely, $N \leftrightarrow I1 \leftrightarrow I2 \leftrightarrow U$ vs $N \leftrightarrow I1' \leftrightarrow U$ and $N \leftrightarrow I2' \leftrightarrow U$), the claim of multiple parallel pathways in this case stands corrected. Alternate routes to prove existence of multiple parallel pathways include measuring the upward curvature of $\log(kf(u))$ vs Force or denaturant concentration in Chevron plots (see for example Wright, Caroline F., et al. "Parallel protein-unfolding pathways revealed and mapped." Nature Structural & Molecular Biology 10.8 (2003): 658-662.) or using multiple single molecule

measurement probes (e.g., Kim, Jae-Yeol, and Hoi Sung Chung. "Disordered proteins follow diverse transition paths as they fold and bind to a partner." *Science* 368.6496 (2020): 1253-1257.).

2. The authors hint (on p. 13 and p. 16) that the origin of templating between RTX-iv and RTX-v could be the inter-facial protein-protein interactions between the two domains. Since an atomic resolution 3D structure of the two domains is available, it might be of interest to the reader to have a paragraph in the discussion section about the structural basis (e.g., specific residue contacts) of the notion of templating and how it might provide a more general mechanism of IDP folding upon binding. The structure examination can then be further extended to compare the interface between v-iv and iv-iii (using PDB 7RAH), because the authors aim to imply a more generalized "transmitting/relaying/successive" template effect that propagate from the C- to N- terminus for the complete folding of RTX blocks. Otherwise, such proposed mechanism would be best supported by actual OT experiments with the full-length RTX construct to show strict hierarchical folding order, from v to i in a stepwise manner.

Minor comments/corrections

1. The central thesis of this manuscript is that folding of RTX-iv is templated by folding of RTX-v. The authors correctly deduce that a strict folding hierarchy between the two domains is an essential pre-requisite for this argument to hold. The sequential unfolding of iv always preceding before v is readily distinguished by the characteristic unfolding forces (10 pN for iv and 13 pN for v) and the change in contour length (ΔLc -RTX-iv of 39.9 Å \pm 0.8 nm and a ΔLc -RTX-v of 46.2 Å \pm 0.9 nm; Fig. 3). The sequential refolding of v always preceding before iv, however, is not as straightforward owing to the nearly identical refolding force (both occur \sim 4 pN). Hence, the authors should highlight/direct readers attention to the WLC fits in Figure 3, provide the respective ΔLc 's observed for the two refolding transitions, and comment whether those ΔLc 's are consistent with the WLC fits in the v-preceding-iv refolding scenario. It is important that the authors clearly explain the protocol of refolding order assignment in the main text. Otherwise, additional data/analysis must be presented to support the strong claim made by the authors that there is indeed no exception to the rule that RTX-v folds before RTX-iv in RTX-iv-v.

2. On p. 11, the authors claim "To rigorously prove the existence of the templated folding of RTX-iv by RTX-v, it is imperative to prove that the folding of RTX-iv is conditional upon the folding of RTX-v. If the templating effect exists, slowing down the folding of RTX-v should also decelerate the folding of RTX-iv." This statement is too strong and misplaced since the fact that folding of RTX-iv is conditional on folding of RTX-v has already been demonstrated by the authors when they observed that RTX-iv folds in tandem with RTX-v, but not in isolation. So, the loop-insertion RTX-v experiment acts as a confirmation, rather than the "proof" of this conditional folding phenomenon.

3. On p. 5, the statement "As shown in Fig. 2A, the far UV circular dichroism (CD) spectra of RTX-iv alone were identical in Ca²⁺-free buffer and buffer containing 10 mM Ca²⁺..." should be changed to "As shown in Fig. 2A, the far UV circular dichroism (CD) spectra of RTX-iv alone were similar in Ca²⁺-free buffer and buffer containing 10 mM Ca²⁺..."

4. On p. 7, Figure 3 caption has several mistakes, including: (i) "RTX-v (colored in green)" should be changed to "RTX-v (colored in cyan)". (ii) "The unfolding and refolding events of NuG2 domains are colored in black and indicated by the triangles." should be changed to "The unfolding and refolding events of NuG2 domains are colored in black and indicated by rectangles." (iii) Caption for panel B) was missing, and the rest are therefore mislabeled (i.e., B should be C, C should be D, D should be E, and accordingly "Grey dashed lines in B) and C) are..."). (iv) What is the equation for a "pseudo WLC" fit? (v) For panel C), the authors should specify that "the stretching-relaxation cycles shown therein were limited to below 11-12 pN so that only RTX-iv unfolds and refolds, while RTX-v remained folded throughout." This strategy is similar but not identical to the pulling

protocol used for data shown in panel B), and hence needs to be clearly explained to avoid confusion.

5. The contour length analysis relies heavily on WLC fitting but nowhere is the WLC equation mentioned, nor an appropriate reference cited.

6. On p. 8, first paragraph, the statement "In comparison, RTX-v unfolded at ~ 13 pN and refolded at ~ 4 pN, which are the same as those of standalone RTX-v (Fig. S1).²⁵" should be clarified as "In comparison, RTX-v unfolded at ~ 13 pN and refolded at ~ 4 pN (Fig. S1), which are the same as those of standalone RTX-v reported in Ref 25." The current statement is misleading because the data in Fig. S1 are taken with a RTX-vi-v construct, not with a standalone RTX-v.

7. On p. 12, last paragraph, the statement "It is important to point out that upon relaxed to zero force" should be changed to "It is important to point out that upon relaxing to zero force"

8. Caption for Fig. 5 has mislabeled panel (E) as (C), and therefore missing legends for panels (B), (C), and (D). For instance, (B) is showing a representative F-D curve, not consecutive curves. (C) and (D) are both showing consecutive curves but (C) with waiting time of 0 sec and (D) with 60 sec in between stretching-relaxation cycles. Also, it should be "the second unfolding group (colored in cyan)" not blue.

9. On p. 13, top, the statement "Similarly, we did not observe any trajectory in which both CF-I and RTX-iv folded" is confusing and should be changed to: "Similarly, we did not observe any trajectory in which both CF-I and RTX-iv refolded." It is certainly true that any force spectroscopy instrument will not be able to detect a conformational transition that only occurs at zero force. The authors can point out this fact and suggest future experiments where pulling rates can be varied to further characterize the details of this refolding pathway.

10. On p. 13, the reference to the Bell-Evans model must be cited and the equation used for fitting the rates as a function of force in Fig S2 must be provided in the supplementary information. Moreover, the method to extract rates from non-equilibrium pulling experiments should be described briefly in the supplementary information.

11. On p. 14, top, the statement "A plausible explanation for this observed peculiar mechanical unfolding hierarchy is that RTX-v is stabilized by the folded RTX-v to the extent that its unfolding force is much higher than that of RTX-iv, consequently the mechanically weaker RTX-iv always unfolds first." should, for clarity, be rewritten as: "A plausible explanation for this observed peculiar mechanical unfolding hierarchy is that RTX-v is stabilized by the folded RTX-iv to the extent that its unfolding force is much higher than that of RTX-iv, consequently the mechanically weaker RTX-iv always unfolds first."

12. On p.15, first paragraph, the statement "Similarly, OT experiments on the variant RTX-iv-F1465C-v revealed similar results, i.e. RTX-v was observed to unfold at ~ 30 pN prior to the folding of RTX-iv-F1465C (Fig. S4)." should be changed to "Likewise, OT experiments on the variant RTX-iv-F1465C-v revealed similar results, i.e. RTX-v was observed to unfold at ~ 30 pN prior to the unfolding of RTX-iv-F1465C (Fig. S4)."

13. On p.15, first paragraph, the authors quantify the stabilization effects of RTX-iv on RTX-v and vice versa without the expressions used to derive them. The explicit calculations of these effects are essential for completeness.

14. In Fig. S5, the authors plot lifetime distributions of RTX-iv-F1465C and RTX-iv-T1483C but do not mention how these were measured, and whether these lifetimes are force dependent. This information should be added.

15. In Table S1, it is unclear why the "Fu" parameter is identical in all reactions and has different significant digits than the "Ff" parameter. Also, how are the errors calculated in this table and in the rest of the tables and figures in the manuscript?

16. On p. 16, bottom, the statement "If this mechanism holds, it would suggest that it will be feasible to abolish toxin activity by directly disrupting the CR3 binding site or allosterically." should be changed to "If this mechanism holds, it would suggest that it will be feasible to abolish toxin activity by directly disrupting the CR3 binding site or allosterically."

Summary

This paper titled “Templated Folding of the RTX Domain of the Bacterial Toxin Adenylate Cyclase Revealed by Single Molecule Force Spectroscopy” is a follow-up work by the authors, who previously had used single-molecule optical tweezers studies to characterize the folding/unfolding and mechanical properties of the C-terminal repeats-in-toxin (RTX) block v domain (residue 1529-1681) from the bacterial toxin adenylate cyclase (CyaA) (Wang *et al*, JACS 2019). Here in this new work, Wang and Li have extended their survey on the two tandem domains, i.e., block iv and v (residue 1377-1681), in the C-terminus of CyaA and systematically interrogated how these two domains modulate each another’s folding/unfolding pathway as well as structural/mechanical stability. Their experimental results indicate that: (i) the two domains, RTX-iv and RTX-v, fold in a strict hierarchical fashion, such that folding of RTX-v always precedes folding of RTX-iv; (ii) folding of RTX-iv is conditional upon the folding of RTX-v, such that introducing an unstructured loop to paralyze folding of the N-terminal part of RTX-v, in turn, prevents folding of RTX-iv; and (iii) folded RTX-iv imparts mechanical strength to RTX-v, such that stretching RTX-v and the C-terminus of RTX-iv while keeping the N-terminus of RTX-iv folded dramatically increases the force (13 to 30 pN) required to unfold RTX-v. Based on these results, the authors propose that the folding of RTX-iv is “templated” by its partner RTX-v, due to interfacial interactions between the two domains. Ultimately, the authors mention that these observations have implications for the development of therapeutics against the pathogenicity of CyaA, a key virulence factor of whooping cough.

Overall, the work of this paper is of sound quality, the layout is clear and easy to comprehend, and the results are of great interest to the protein folding field, and with implications on how the folding of CyaA toxin can be specifically impaired as one potential therapeutic approach. We would recommend this paper for publication in Nature Communications after the authors address the following issues and complete the modifications/edits as suggested below:

Major Comments

1. The authors indicated, based on their WLC fits to unfolding and refolding traces of the RTX-iv-v construct, that they observe both, only one, or none of the two intermediates along the transition paths of RTX-iv. Based on this observation, they suggest that RTX-iv folds and unfolds via multiple parallel pathways: $N \leftrightarrow I1 \leftrightarrow I2 \leftrightarrow U$, $N \leftrightarrow I1 \leftrightarrow U$, $N \leftrightarrow I2 \leftrightarrow U$, or $N \leftrightarrow U$. An alternative explanation of this observation could be that there is only one unique reversible pathway for folding and unfolding of RTX-iv (i.e. $N \leftrightarrow I1 \leftrightarrow I2 \leftrightarrow U$) but, since the temporal resolution of the optical tweezers experiments is finite, the lifetime of one (or both) of the intermediate states, I1/I2, in certain trajectories may fall shorter than the detection limit (say ~10 ms) and therefore the observed transition path appears to have fewer intermediates in a small minority of cases. This scenario is - equally plausible as the multiple parallel pathways the authors proposed and - consistent with the observation that the respective ΔLc 's remain the same for I1 and I2 in the four-state as well as three-state dataset. The authors should include in the supplementary the representative F-D curves for these “three-state pathways” to assist this discussion. Unless the authors can further characterize these intermediate states along the (un)folding pathway of RTX-iv that differ significantly from I1, I2 observed in the most common 4-state pathway (namely, $N \leftrightarrow I1 \leftrightarrow I2 \leftrightarrow U$ vs $N \leftrightarrow I1' \leftrightarrow U$ and $N \leftrightarrow I2' \leftrightarrow U$), the claim of multiple parallel pathways in this case stands corrected. Alternate routes to prove

existence of multiple parallel pathways include measuring the upward curvature of $\log(k_{f(u)})$ vs Force or denaturant concentration in Chevron plots (see for example Wright, Caroline F., et al. "Parallel protein-unfolding pathways revealed and mapped." *Nature Structural & Molecular Biology* 10.8 (2003): 658-662.) or using multiple single molecule measurement probes (e.g., Kim, Jae-Yeol, and Hoi Sung Chung. "Disordered proteins follow diverse transition paths as they fold and bind to a partner." *Science* 368.6496 (2020): 1253-1257.).

2. The authors hint (on p. 13 and p. 16) that the origin of templating between RTX-iv and RTX-v could be the inter-facial protein-protein interactions between the two domains. Since an atomic resolution 3D structure of the two domains is available, it might be of interest to the reader to have a paragraph in the discussion section about the structural basis (e.g., specific residue contacts) of the notion of templating and how it might provide a more general mechanism of IDP folding upon binding. The structure examination can then be further extended to compare the interface between v-iv and iv-iii (using PDB 7RAH), because the authors aim to imply a more generalized "transmitting/relaying/successive" template effect that propagate from the C- to N-terminus for the complete folding of RTX blocks. Otherwise, such proposed mechanism would be best supported by actual OT experiments with the full-length RTX construct to show strict hierarchical folding order, from v to i in a stepwise manner.

Minor comments/corrections

1. The central thesis of this manuscript is that folding of RTX-iv is templated by folding of RTX-v. The authors correctly deduce that a strict folding hierarchy between the two domains is an essential pre-requisite for this argument to hold. The sequential unfolding of iv always preceding before v is readily distinguished by the characteristic unfolding forces (10 pN for iv and 13 pN for v) and the change in contour length (ΔLc -RTX-iv of 39.9 Å \pm 0.8 nm and a ΔLc -RTX-v of 46.2 Å \pm 0.9 nm; Fig. 3). The sequential refolding of v always preceding before iv, however, is not as straightforward owing to the nearly identical refolding force (both occur \sim 4 pN). Hence, the authors should highlight/direct readers attention to the WLC fits in Figure 3, provide the respective ΔLc 's observed for the two refolding transitions, and comment whether those ΔLc 's are consistent with the WLC fits in the v-preceding-iv refolding scenario. It is important that the authors clearly explain the protocol of refolding order assignment in the main text. Otherwise, additional data/analysis must be presented to support the strong claim made by the authors that there is indeed no exception to the rule that RTX-v folds before RTX-iv in RTX-iv-v.
2. On p. 11, the authors claim "To rigorously prove the existence of the templated folding of RTX-iv by RTX-v, it is imperative to prove that the folding of RTX-iv is conditional upon the folding of RTX-v. If the templating effect exists, slowing down the folding of RTX-v should also decelerate the folding of RTX-iv." This statement is too strong and misplaced since the fact that folding of RTX-iv is conditional on folding of RTX-v has already been demonstrated by the authors when they observed that RTX-iv folds in tandem with RTX-v, but not in isolation. So, the loop-insertion RTX-v experiment acts as a confirmation, rather than the "proof" of this conditional folding phenomenon.

3. On p. 5, the statement “As shown in Fig. 2A, the far UV circular dichroism (CD) spectra of RTX-iv alone were identical in Ca²⁺-free buffer and buffer containing 10 mM Ca²⁺ ...” should be changed to “As shown in Fig. 2A, the far UV circular dichroism (CD) spectra of RTX-iv alone were similar in Ca²⁺-free buffer and buffer containing 10 mM Ca²⁺ ...”
4. On p. 7, Figure 3 caption has several mistakes, including: (i) “RTX-v (colored in green)” should be changed to “RTX-v (colored in cyan)”. (ii) “The unfolding and refolding events of NuG2 domains are colored in black and indicated by the triangles.” should be changed to “The unfolding and refolding events of NuG2 domains are colored in black and indicated by rectangles.” (iii) Caption for panel B) was missing, and the rest are therefore mislabeled (i.e., B should be C, C should be D, D should be E, and accordingly “Grey dashed lines in B) and C) are...”). (iv) What is the equation for a “pseudo WLC” fit? (v) For panel C), the authors should specify that “the stretching-relaxation cycles shown therein were limited to below 11-12 pN so that only RTX-iv unfolds and refolds, while RTX-v remained folded throughout.” This strategy is similar but not identical to the pulling protocol used for data shown in panel B), and hence needs to be clearly explained to avoid confusion.
5. The contour length analysis relies heavily on WLC fitting but nowhere is the WLC equation mentioned, nor an appropriate reference cited.
6. On p. 8, first paragraph, the statement “In comparison, RTX-v unfolded at ~13 pN and refolded at ~4 pN, which are the same as those of standalone RTX-v (Fig. S1).²⁵” should be clarified as “In comparison, RTX-v unfolded at ~13 pN and refolded at ~4 pN (Fig. S1), which are the same as those of standalone RTX-v reported in Ref 25.” The current statement is misleading because the data in Fig. S1 are taken with a RTX-vi-v construct, not with a standalone RTX-v.
7. On p. 12, last paragraph, the statement “It is important to point out that upon relaxed to zero force” should be changed to “It is important to point out that upon relaxing to zero force”
8. Caption for Fig. 5 has mislabeled panel (E) as (C), and therefore missing legends for panels (B), (C), and (D). For instance, (B) is showing a representative F-D curve, not consecutive curves. (C) and (D) are both showing consecutive curves but (C) with waiting time of 0 sec and (D) with 60 sec in between stretching-relaxation cycles. Also, it should be “the second unfolding group (colored in cyan)” not blue.
9. On p. 13, top, the statement “Similarly, we did not observe any trajectory in which both CF-I and RTX-iv folded” is confusing and should be changed to: “Similarly, we did not observe any trajectory in which both CF-I and RTX-iv refolded.” It is certainly true that any force spectroscopy instrument will not be able to detect a conformational transition that only occurs at zero force. The authors can point out this fact and suggest future experiments where pulling rates can be varied to further characterize the details of this refolding pathway.
10. On p. 13, the reference to the Bell-Evans model must be cited and the equation used for fitting the rates as a function of force in Fig S2 must be provided in the supplementary

information. Moreover, the method to extract rates from non-equilibrium pulling experiments should be described briefly in the supplementary information.

11. On p. 14, top, the statement “A plausible explanation for this observed peculiar mechanical unfolding hierarchy is that RTX-v is stabilized by the folded RTX-v to the extent that its unfolding force is much higher than that of RTX-iv, consequently the mechanically weaker RTX-iv always unfolds first.” should, for clarity, be rewritten as: “A plausible explanation for this observed peculiar mechanical unfolding hierarchy is that RTX-v is stabilized by the folded RTX-iv to the extent that its unfolding force is much higher than that of RTX-iv, consequently the mechanically weaker RTX-iv always unfolds first.”
12. On p.15, first paragraph, the statement “Similarly, OT experiments on the variant RTX-iv-F1465C-v revealed similar results, i.e. RTX-v was observed to unfold at ~30 pN prior to the folding of RTX-iv-F1465C (Fig. S4).” should be changed to “Likewise, OT experiments on the variant RTX-iv-F1465C-v revealed similar results, i.e. RTX-v was observed to unfold at ~30 pN prior to the unfolding of RTX-iv-F1465C (Fig. S4).”
13. On p.15, first paragraph, the authors quantify the stabilization effects of RTX-iv on RTX-v and *vice versa* without the expressions used to derive them. The explicit calculations of these effects are essential for completeness.
14. In Fig. S5, the authors plot lifetime distributions of RTX-iv-F1465C and RTX-iv-T1483C but do not mention how these were measured, and whether these lifetimes are force dependent. This information should be added.
15. In Table S1, it is unclear why the “ F_u ” parameter is identical in all reactions and has different significant digits than the “ F_f ” parameter. Also, how are the errors calculated in this table and in the rest of the tables and figures in the manuscript?
16. On p. 16, bottom, the statement “If this mechanism holds, it would suggest that it will be feasibly to abolish toxin activity by directly disrupting the CR3 binding site or allosterically.” should be changed to “If this mechanism holds, it would suggest that it will be feasible to abolish toxin activity by directly disrupting the CR3 binding site or allosterically.”

Reviewer #2 (Remarks to the Author):

In this paper, Wang and Li use single molecule optical tweezers to investigate the calcium-triggered folding of two RTX domains from the bacterial toxin adenylate cyclase (CyaA) of *B. pertussis*. In their prior study (JACS 2019), they investigated the mechanical unfolding–folding properties of the Block V RTX domain of CyaA. Here, they characterize the adjacent Block IV as well as the combined Block IV - Block V polypeptide. They show that RTX Block IV alone is intrinsically disordered even in the presence of excess calcium, while it efficiently folds into a calcium-loaded β -roll structure in the combined Block IV/V protein. They provide evidence that RTX-iv and RTX-v are two folded thermodynamic entities and that folding of RTX-IV is dependent upon the folding of RTX-V. Conversely, the folded RTX-IV has a clear stabilization effect on RTX-V. They propose that this mutual stabilization may contribute to the efficient secretion of the RTX domain through the Type I secretion system.

This is a good set of experiments that provide interesting data on the folding mechanisms of the RTX proteins. Yet, this study appears mainly as an extension of the prior work of Wang and Li and somehow lacks significant novelty or groundbreaking insights into the secretion mechanisms. Numerous structural and biophysical studies have previously documented the critical role of capping structures for the folding of RTX β -rolls. The present single molecule study fully corroborates the well-established calcium-triggered folding of RTX proteins without bringing truly new concepts.

Major comments:

1. In their prior work (JACS 2019) , the authors reported that the RTX-V unfolded following a three-state pathway involving the formation of an unfolding intermediate state, while in the present study RTX-V in the RTX-IV/V polypeptide seems to follow a two-state pathway (in Fig. 3 & 4). What could be the reason for this difference ?
2. In Fig. 5 , the authors characterized a variant RTX-v50 carrying a 50-aa loop inserted between two consecutive β -strands. This insertion has a pronounced effect on the folding kinetics of RTX-v but apparently not on the folding itself, which is quite striking. It would be nice to present independent biophysical evidences (e.g. CD) documenting the proper folding of this recombinant protein.
3. All experiments have been carried out in the presence of a single (and high) calcium concentration. Actually, varying the calcium concentration and/or the ionic strength could be a much more subtle approach to control the folding kinetics of the proteins. Have the authors explored the unfolding–folding properties at different (and more physiological) calcium concentrations.
4. I do not fully buy the reversed unfolding hierarchy between RTX-IV and V in the Fig. 6 experiments. Could the experimental data be explained or interpreted otherwise in particular considering the three-state unfolding pathway of RTX-V proposed previously (see above , JACS 2019) ?
5. In these Fig. 6 experiments, the unfolding of RTX-v occurred at higher forces (~ 30 pN) reaching those triggering unfolding of the NuG2 modules (although this is not apparent on the traces). How did the authors cope with this potential problem ?

Minor comment:

1. The legend of Fig. 3 is inaccurate.

Reviewer #3 (Remarks to the Author):

Wang and Li have studied an aspect of the folding of adenylate cyclase toxin that is relevant

to all bacterial proteins that are exported via the Type I Secretion System. These are RTX (Repeats-in-Toxin) proteins, which are secreted in an unfolded state C-terminal end first and begin to fold in the Ca²⁺-rich extracellular medium. The RTX repeats of this model protein are in five blocks (i-v). The C-terminal block (RTX-v) is the first to fold, and here the authors have investigated how its folding is propagated to the next block of RTX repeats (RTX-iv). What is remarkable about this discovery is that RTX-iv expressed on its own is completely unstructured both in the absence of Ca²⁺ and in the presence of 10 mM Ca²⁺, as shown here by CD. The crystal structure of these two domain blocks (iv and v) suggests that there is an end-to-end association that might propagate the RTX fold from v into iv. Here the authors have used single molecule optical tweezers and mutagenesis to show that the folding of RTX-v transmits the folding signal into the next domain, and that without this RTX-iv would remain unstructured. This seeding process may well be transmitted to upstream RTX block and even beyond into other Ca²⁺-dependent folds. The authors have done a thorough job in probing the transmission of the folding from RTX-v to RTX-iv using the optical tweezers technique. They have moved from the crystal structure endpoint into a solution technique that has dissected the process of transmission of the fold. They have made strategic Cys mutations as tweezer attachment points in different parts of the protein to dissect the process. This strategy could well be useful to study folding beyond RTX-iv in the direction of the N terminus and in other RTX proteins. The authors mention the possibility of using these results to develop new therapeutics to combat *Bacillus pertussis*. This sounds rather vague and it would be helpful to have a bit more information on how this might be achieved.

Corrections

The legend to Figure 3 seems to address a different version of this figure with one less panel. There is no text for panel E), and D) is described as the histogram, which is panel E). Panel A is supposed to have an inset, which is missing. Also, the plot said to be green appears to be cyan to me.

Accessibility. The issue with colours being used as the only distinguishing feature in a figure is the problem readers might have with colour-blindness. To counter this one set of histograms in Figure 3E could be cross hatched in addition to being a different colour. In Figure 2A the three lines could also be distinguished by being solid, dotted, or dashed in addition to the colour difference.

The green asterisks that mark the Cys mutations in RTX-iv in Figures 4A and 6A should be made larger to stand out more – perhaps twice the diameter.

The manuscript is well written and clearly described. However, there are few editorial points to make.

Tris buffers should always stipulate the other buffer component, which is HCl most of the time viz. Tris-HCl.

Page 7, the Figure 3 reference should come a little earlier in the paragraph.

Page 9, three lines down: 'are resulted' would be better as – 'are derived'. Ditto on Page 8, first paragraph.

Page 15, first line of Discussion: critical roles.

Page 16, 'and so on so forth' should be: 'and so on and so forth'. RTX-I should be RTX-i. '...disrupting the CR3 binding site or allosterically.' needs clarification/rewording.

Page 17, 5' and 3' should be written with a prime rather than an apostrophe.

Occasionally an article is missing as on page 18, 10 lines down: the C-terminus.

On Page 19, a sentence starts with a numeral (1 μ L), which is to be avoided.

In Acknowledgements the NSERC DG number should be supplied.

Many thanks for reviewing our manuscript (MS# NCOMMS-21- 21-43970) entitled “Templated Folding of the RTX Domain of the Bacterial Toxin Adenylate Cyclase Revealed by Single Molecule Force Spectroscopy”. We thank all reviewers for their enthusiasm in our manuscript and constructive suggestions/comments. Following these comments/suggestions, we revised our manuscript accordingly. Changes we made to address reviewers’ comments are highlighted in yellow in the revised manuscript. We hope that our manuscript is now acceptable for publication in *Nature Communications*.

The response to the reviewer’s comments is detailed as follows:

Reviewer #1

1. “The authors indicated, based on their WLC fits to unfolding and refolding traces of the RTX-iv-v construct, that they observe both, only one, or none of the two intermediates along the transition paths of RTX-iv. Based on this observation, they suggest that RTX-iv folds and unfolds via multiple parallel pathways: $N \leftrightarrow I1 \leftrightarrow I2 \leftrightarrow U$, $N \leftrightarrow I1 \leftrightarrow U$, $N \leftrightarrow I2 \leftrightarrow U$, or $N \leftrightarrow U$. An alternative explanation of this observation could be that there is only one unique reversible pathway for folding and unfolding of RTX-iv (i.e. $N \leftrightarrow I1 \leftrightarrow I2 \leftrightarrow U$) but, since the temporal resolution of the optical tweezers experiments is finite, the lifetime of one (or both) of the intermediate states, I1/I2, in certain trajectories may fall shorter than the detection limit (say ~10 ms) and therefore the observed transition path appears to have fewer intermediates in a small minority of cases. This scenario is - equally plausible as the multiple parallel pathways the authors proposed and - consistent with the observation that the respective ΔLc ’s remain the same for I1 and I2 in the four-state as well as three-state dataset. The authors should include in the supplementary the representative F-D curves for these “three-state pathways” to assist this discussion.

Unless the authors can further characterize these intermediate states along the (un)folding pathway of RTX-iv that differ significantly from I1, I2 observed in the most common 4-state pathway (namely, $N \leftrightarrow I1 \leftrightarrow I2 \leftrightarrow U$ vs $N \leftrightarrow I1' \leftrightarrow U$ and $N \leftrightarrow I2' \leftrightarrow U$), the claim of multiple parallel pathways in this case stands corrected. Alternate routes to prove existence of multiple parallel pathways include measuring the upward curvature of $\log(kf(u))$ vs Force or denaturant concentration in Chevron plots (see for example Wright, Caroline F., et al. "Parallel protein-unfolding pathways revealed and mapped." *Nature Structural & Molecular Biology* 10.8 (2003): 658-662.) or using multiple single molecule measurement probes (e.g., Kim, Jae-Yeol, and Hoi Sung Chung. "Disordered proteins follow diverse transition paths as they fold and bind to a partner." *Science* 368.6496 (2020): 1253-1257).”

Response: we appreciate this insightful comment. Following this suggestion, we have included representative *F-D* curves showing the “three-state” pathways of RTX-iv in the revised Supplementary Information as new Fig. S2. We agree with this reviewer that it is plausible that the three-state unfolding trajectories, in which either II or I2 is missing, were due to the limited temporal resolution of the OT. We thank this reviewer for pointing out alternative methods to prove the existence of the multiple parallel pathways. Limited by the temporal resolution of our OT setup and also due to the low occurrence of these “three-state” pathways (~3.7%), we are not able to characterize these pathways in more detail to discern if these intermediate states differ significantly from I1 and I2 observed in 4-state pathway. Therefore, in the revised manuscript we have acknowledged that although it is possible that these different pathways indicate the existence of parallel (un)folding pathways, it is also plausible that there exists only one unique four-state (un)folding pathway and the apparent three-state pathways may arise due to the limited temporal resolution of our OT measurements (see page 10).

2. *“The authors hint (on p. 13 and p. 16) that the origin of templating between RTX-iv and RTX-v could be the inter-facial protein-protein interactions between the two domains. Since an atomic resolution 3D structure of the two domains is available, it might be of interest to the reader to have a paragraph in the discussion section about the structural basis (e.g., specific residue contacts) of the notion of templating and how it might provide a more general mechanism of IDP folding upon binding. The structure examination can then be further extended to compare the interface between v-iv and iv-iii (using PDB 7RAH), because the authors aim to imply a more generalized “transmitting/relaying/successive” template effect that propagate from the C- to N- terminus for the complete folding of RTX blocks. Otherwise, such proposed mechanism would be best supported by actual OT experiments with the full-length RTX construct to show strict hierarchical folding order, from v to i in a stepwise manner.”*

Response: we appreciate this comment. Indeed, understanding the origin of the templating effect between RTX-iv and RTX-v (and potentially between other neighboring RTX blocks) is a major task in the field. Our current work, together with the pioneering work by Sebo et al^{1,2}, is the first step to demonstrate the existence of the templating effect. Although crystal structures of RTX-v, RTX-iv-v and RTX-i-iii fused with an RTX-v capping structure are available, it remains unknown how this templating effect works and what residues are responsible for this effect. To answer these questions, more in-depth protein engineering work will be required. This work is beyond the scope of this study and will be pursued in our future efforts. It is of note that the structure of the N-terminal part of the RTX-iv is missing in the crystal structure of RTX-iv-v (PDB 5SUS) and RTX-i-iii (PDB 7RAH), thus it is not feasible to speculate what residues located at the interfaces of neighboring RTX blocks are critical for the template effect. We agree with this reviewer that without experimental data on the full length RTX-i-v or identification of critical residues responsible for the templating effect, it is premature to try to extrapolate the observed templating effect in RTX-iv-v to RTX-i-v. Therefore, we have removed our hypothesis on a more generalized “transmitting/relaying/successive” template effect from our revised manuscript.

Minor comments/corrections

1. *“The central thesis of this manuscript is that folding of RTX-iv is templated by folding of RTX-*

v. The authors correctly deduce that a strict folding hierarchy between the two domains is an essential pre-requisite for this argument to hold. The sequential unfolding of iv always preceding before v is readily distinguished by the characteristic unfolding forces (10 pN for iv and 13 pN for v) and the change in contour length (ΔLc -RTX-iv of $39.9 \text{ \AA} \pm 0.8 \text{ nm}$ and a ΔLc -RTX-v of $46.2 \text{ \AA} \pm 0.9 \text{ nm}$; Fig. 3). The sequential refolding of v always preceding before iv, however, is not as straightforward owing to the nearly identical refolding force (both occur $\sim 4 \text{ pN}$). Hence, the authors should highlight/direct readers attention to the WLC fits in Figure 3, provide the respective ΔLc 's observed for the two refolding transitions, and comment whether those ΔLc 's are consistent with the WLC fits in the v-preceding-iv refolding scenario. It is important that the authors clearly explain the protocol of refolding order assignment in the main text. Otherwise, additional data/analysis must be presented to support the strong claim made by the authors that there is indeed no exception to the rule that RTX-v folds before RTX-iv in RTX-iv-v."

Response: following this helpful suggestion, we have now clearly labelled the contour length increment ΔLc for the folding events of RTX-iv and RTX-v in Fig. 3. The assignment of folding events of RTX-iv and RTX-v was based on ΔLc and pseudo WLC fits. The pseudo WLC fits clearly showed that the refolding event, which occurred first, during relaxation had a larger ΔLc than the second refolding event, indicating that RTX-v refolded first. We also used consecutive stretching-relaxation curves to further verify this assignment (as shown in the new Fig. S3). After RTX-iv-v had unfolded in the stretching curve, we relaxed the molecule until the first folding event occurred. Then the molecule was stretched again to unfold the refolded RTX block. From this subsequent unfolding event (which occurs at $\sim 9 \text{ pN}$), we can accurately determine its ΔLc and assign the first refolding event to the refolding of RTX-v. We have now included these details in the revised manuscript (see page 8 and 11).

2. *"On p. 11, the authors claim "To rigorously prove the existence of the templated folding of RTX-iv by RTX-v, it is imperative to prove that the folding of RTX-iv is conditional upon the folding of RTX-v. If the templating effect exists, slowing down the folding of RTX-v should also decelerate the folding of RTX-iv." This statement is too strong and misplaced since the fact that folding of RTX-iv is conditional on folding of RTX-v has already been demonstrated by the authors when they observed that RTX-iv folds in tandem with RTX-v, but not in isolation. So, the loop-insertion RTX-v experiment acts as a confirmation, rather than the "proof" of this conditional folding phenomenon."*

Response: we concur with this reviewer that the original statement is too strong. Therefore, we have revised this statement to "To further corroborate the existence of the templated folding of RTX-iv by RTX-v, we endeavored to prove that the folding of RTX-iv is conditional upon the folding of RTX-v." (see page 11-12 in the revised manuscript).

3. *"On p. 5, the statement "As shown in Fig. 2A, the far UV circular dichroism (CD) spectra of RTX-iv alone were identical in Ca^{2+} -free buffer and buffer containing 10 mM Ca^{2+} ..." should be changed to "As shown in Fig. 2A, the far UV circular dichroism (CD) spectra of RTX-iv alone were similar in Ca^{2+} -free buffer and buffer containing 10 mM Ca^{2+} ..."*

Response: we have made the suggested change (see page 5 in the revised manuscript).

4. *“On p. 7, Figure 3 caption has several mistakes, including: (i) “RTX-v (colored in green)” should be changed to “RTX-v (colored in cyan)”. (ii) “The unfolding and refolding events of NuG2 domains are colored in black and indicated by the triangles.” should be changed to “The unfolding and refolding events of NuG2 domains are colored in black and indicated by rectangles.” (iii) Caption for panel B) was missing, and the rest are therefore mislabeled (i.e., B should be C, C should be D, D should be E, and accordingly “Grey dashed lines in B) and C) are...”. (iv) What is the equation for a “pseudo WLC” fit? (v) For panel C), the authors should specify that “the stretching-relaxation cycles shown therein were limited to below 11-12 pN so that only RTX-iv unfolds and refolds, while RTX-v remained folded throughout.” This strategy is similar but not identical to the pulling protocol used for data shown in panel B), and hence needs to be clearly explained to avoid confusion.”*

Response: we regret these mistakes. We have now corrected these mistakes in the revised manuscript. With regard to pseudo WLC fits: we used WLC model of polymer elasticity to fit the F-D curves. WLC model describes the force-extension relationship of a polymer chain, and cannot be used to fit F-D curves. Using WLC model to fit F-D curves will not yield meaningful persistence length or ΔLc , therefore we call this fitting as pseudo WLC fits. Although pseudo fits do not lead to the accurate determination of ΔLc , these fits can still serve as rough guides allowing us to distinguish different (un)folding events. Accurate ΔLc are obtained from Fig. 3B. We have explained these points in detail in the revised manuscript (see page 8 and 21-22). In addition, in the legend of Fig. 3, we have also specified that the stretching-relaxation curves shown in Fig. 3C were limited to below 11-12 pN so that only RTX-iv unfolds and refolds, while RTX-v remained folded throughout (see page 8 in the revised manuscript).

5. *“The contour length analysis relies heavily on WLC fitting but nowhere is the WLC equation mentioned, nor an appropriate reference cited.”*

Response: we have now cited the original WLC paper (Marko and Siggia, *Macromolecules*, 1995) and included the WLC equation in the revised manuscript (see new Ref. 35 and page 21).

6. *“On p. 8, first paragraph, the statement “In comparison, RTX-v unfolded at ~13 pN and refolded at ~4 pN, which are the same as those of standalone RTX-v (Fig. S1).25” should be clarified as “In comparison, RTX-v unfolded at ~13 pN and refolded at ~4 pN (Fig. S1), which are the same as those of standalone RTX-v reported in Ref 25.” The current statement is misleading because the data in Fig. S1 are taken with a RTX-vi-v construct, not with a standalone RTX-v.”*

Response: we have revised this statement to make this point clear (see page 8).

7. *“On p. 12, last paragraph, the statement “It is important to point out that upon relaxed to zero force” should be changed to “It is important to point out that upon relaxing to zero force”*

8. *Caption for Fig. 5 has mislabeled panel (E) as (C), and therefore missing legends for panels (B), (C), and (D). For instance, (B) is showing a representative F-D curve, not consecutive curves. (C) and (D) are both showing consecutive curves but (C) with waiting time of 0 sec and*

(D) with 60 sec in between stretching-relaxation cycles. Also, it should be “the second unfolding group (colored in cyan)” not blue.

Response: we have corrected these typos/mistakes in the labelling (see page 12 and 14 in the revised manuscript).

9. “On p. 13, top, the statement “Similarly, we did not observe any trajectory in which both CF-I and RTX-iv folded” is confusing and should be changed to: “Similarly, we did not observe any trajectory in which both CF-I and RTX-iv refolded.” It is certainly true that any force spectroscopy instrument will not be able to detect a conformational transition that only occurs at zero force. The authors can point out this fact and suggest future experiments where pulling rates can be varied to further characterize the details of this refolding pathway.

Response: following these helpful suggestions, we have made corresponding changes (see page 14 in the revised manuscript). Due to the slow folding kinetics of the N-terminal part of RTX-v₅₀, no refolding events were observed in the relaxation curves even at a pulling rate of 2 nm/s. Therefore, we used an approach that is similar to the double pulse approach used in the AFM-based force spectroscopy experiments to characterize the folding kinetics of proteins. We have cited relevant references^{3,4} to make this point clear (see page 14).

10. “On p. 13, the reference to the Bell-Evans model must be cited and the equation used for fitting the rates as a function of force in Fig S2 must be provided in the supplementary information. Moreover, the method to extract rates from non-equilibrium pulling experiments should be described briefly in the supplementary information.”

Response: following these suggestions, we have now cited relevant papers for the Bell-Evans model and included the equations for fitting the rate vs force curves (see legend of Fig. S5 in Supplementary Information). In addition, we also included a short description of the Oesterhelt method we used to extract rates from non-equilibrium pulling experiments (see page 22 in revised manuscript).

11. “On p. 14, top, the statement “A plausible explanation for this observed peculiar mechanical unfolding hierarchy is that RTX-v is stabilized by the folded RTX-v to the extent that its unfolding force is much higher than that of RTX-iv, consequently the mechanically weaker RTX-iv always unfolds first.” should, for clarity, be rewritten as: “A plausible explanation for this observed peculiar mechanical unfolding hierarchy is that RTX-v is stabilized by the folded RTX-iv to the extent that its unfolding force is much higher than that of RTX-iv, consequently the mechanically weaker RTX-iv always unfolds first.”

12. On p.15, first paragraph, the statement “Similarly, OT experiments on the variant RTX-iv-F1465C-v revealed similar results, i.e. RTX-v was observed to unfold at ~30 pN prior to the folding of RTX-iv-F1465C (Fig. S4).” should be changed to “Likewise, OT experiments on the variant RTX-iv-F1465C-v revealed similar results, i.e. RTX-v was observed to unfold at ~30 pN prior to the unfolding of RTX-iv-F1465C (Fig. S4).”

Response: following these helpful suggestions in Comment 11-12, we have revised manuscript accordingly (see page 15 and 16).

13. *“On p.15, first paragraph, the authors quantify the stabilization effects of RTX-iv on RTX-v and vice versa without the expressions used to derive them. The explicit calculations of these effects are essential for completeness”*

Response: we have included the equations used to quantify the stabilization effect in the revised manuscript (see page 17).

14. *“In Fig. S5, the authors plot lifetime distributions of RTX-iv-F1465C and RTX-iv-T1483C but do not mention how these were measured, and whether these lifetimes are force dependent. This information should be added.”*

Response: we appreciate this comment. We have now included a new figure (new Fig. S8) to show how the lifetime of RTX-iv was measured. Since the lifetime of RTX-iv is short in the F-D curves of RTX-iv-F1465C and RTX-iv-T1483C, we assumed that the force was constant during the lifetime τ . These lifetimes are indeed force dependent, allowing us to estimate the unfold rate constant of RTX-iv at zero force in the absence of a folded RTX-v. We have now included this discussion in the revised manuscript (see page 17) and Supplementary Information (new Fig. S8).

15. *“In Table S1, it is unclear why the “Fu” parameter is identical in all reactions and has different significant digits than the “Ff” parameter. Also, how are the errors calculated in this table and in the rest of the tables and figures in the manuscript?”*

Response: we regret the oversight. We have now corrected these errors (see Table S1 in Supplementary Information). The standard deviation (S.D.) of F_u and F_f was calculated directly from the raw data, and the S.D. of the rate constants were obtained from the fitting. We have now included this information in the revised manuscript (see the legend of Table S1 in Supplementary Information).

16. *On p. 16, bottom, the statement “If this mechanism holds, it would suggest that it will be feasibly to abolish toxin activity by directly disrupting the CR3 binding site or allosterically.” should be changed to “If this mechanism holds, it would suggest that it will be feasible to abolish toxin activity by directly disrupting the CR3 binding site or allosterically.”*

Response: to address this reviewer’s comment #2, we have removed the hypothesis that templating is a general mechanism in the whole RTX domain and its implication in the revised manuscript. As a result, this statement has been removed from the revised manuscript.

Reviewer#2

Major comments:

1. *‘In their prior work (JACS 2019) , the authors reported that the RTX-V unfolded following a three-state pathway involving the formation of an unfolding intermediate state, while in the*

present study RTX-V in the RTX-IV/V polypeptide seems to follow a two-state pathway (in Fig. 3 & 4). What could be the reason for this difference?"

Response: in our prior work, we reported that the unfolding of RTX-v follows two parallel pathways, one is two-state and the other is three-state pathway. In the construct RTX-iv-v, RTX-v unfolded following the same fashion as in our prior work, i.e. both two-state and three-state unfolding pathways. In Fig. 3B, RTX-v unfolded in a two-state fashion in the curve on the right, but unfolded in a three-state fashion in the curve on the left. Due to the short lifetime of the intermediate state and small size of the figure, the three-state unfolding of RTX-v is not too obvious. To make this point clear, we have included the following sentence "RTX-v unfolded at ~13 pN and refolded at ~4 pN (Fig. S1), and its unfolding and folding showed both two-state and three-state behaviors. These features are the same as those of standalone RTX-v reported previously in Ref.⁵" (see page 8 in the revised manuscript).

2. "In fig. 5 , the authors characterized a variant RTX-v50 carrying a 50-aa loop inserted between two consecutive β -strands. This insertion has a pronounced effect on the folding kinetics of RTX-v but apparently not on the folding itself, which is quite striking. It would be nice to present independent biophysical evidences (e.g. CD) documenting the proper folding of this recombinant protein."

Response: following this helpful suggestion, we have now included CD spectra of both RTX-v50 and RTX-iv-RTX-v50 as new Fig. S4 in the revised Supplementary information. In the presence of 10 mM Ca²⁺, both spectra indicated that both proteins are folded. We have included a short paragraph to make this point clear (also see page 13).

3. "All experiments have been carried out in the presence of a single (and high) calcium concentration. Actually, varying the calcium concentration and/or the ionic strength could be a much more subtle approach to control the folding kinetics of the proteins. Have the authors explored the unfolding–folding properties at different (and more physiological) calcium concentrations."

Response: we appreciate this comment. In this work, we carried out all our experiments in the presence of 10 mM Ca²⁺, which is the super-physiological Ca²⁺ concentration, and did not vary Ca²⁺ concentration or ionic strength. We agree with this reviewer that varying the calcium concentration and/or the ionic strength could effectively control the folding kinetics of the proteins, and we have now indicated that we will conduct these studies in future to further characterize the biophysical properties of RTX-iv-v (see page 11).

4. I do not fully buy the reversed unfolding hierarchy between RTX-IV and V in the Fig. 6 experiments. Could the experimental data be explained or interpreted otherwise in particular considering the three-state unfolding pathway of RTX-V proposed previously (see above , JACS 2019) ?

Response: it is well established that in force spectroscopy experiments, contour length increments (ΔL_c) provide unambiguous fingerprints for identifying/assigning (un)folding events. In this case, the unfolding events in Fig. 6B at ~30 pN showed a ΔL_c of 45 nm, which is the

unfolding signature of the standalone RTX-v. In addition, during refolding, the refolding event occurred first showed a ΔLc of 45 nm, again corroborating that this event originates from RTX-v.

5. *In these Fig. 6 experiments, the unfolding of RTX-v occurred at higher forces (~30 pN) reaching those triggering unfolding of the NuG2 modules (although this is not apparent on the traces). How did the authors cope with this potential problem ?*

Response: the two constructs used in Fig. 6 and Fig. S4 do not contain NuG2 domain, so that the unfolding of NuG2 domain will not interfere with those of RTX-v. We have now made this point clear in the revised manuscript (see page 16).

Minor comment:

1. *The legend of Fig. 3 is inaccurate.*

Response: we regret the errors we made in the figure legend of Fig. 3 due to the changes of the version of the figure. We have now corrected these errors (also see our response to Comment 15 of Reviewer#1).

Reviewer#3

1. *“The authors mention the possibility of using these results to develop new therapeutics to combat Bacillus pertussis. This sounds rather vague and it would be helpful to have a bit more information on how this might be achieved”.*

Response: we appreciate this comment. If the templating effect is valid in the folding of the full length RTX, it can be envisioned that it is possible to abolish the toxin activity *by disrupting the CR3 binding site allosterically, i.e. by disrupting the C-terminal RTX blocks* (such as RTX-v or RTX-iv) to disrupt the CR3 binding site, which is located in the interface between RTX-ii and RTX-iii. However, since we do not know the exact molecular mechanism for the templating effect between RTX-iv and RTX-v and we do not have experimental evidence to support that hypothesis that templating effect is a general mechanism in the full length RTX domain, it is premature to extrapolate the templating effect to the whole RTX domain. We have removed the claim of this hypothesis from the revised manuscript (also see our response to Comment #2 of Reviewer #1).

Corrections

1. *“The legend to Figure 3 seems to address a different version of this figure with one less panel. There is no text for panel E), and D) is described as the histogram, which is panel E). Panel A is supposed to have an inset, which is missing. Also, the plot said to be green appears to be cyan to me.”*

Response: we regret for these oversights. Indeed, due to the version change of Fig. 3, we failed to update the figure legend. We have now corrected these errors.

2. *“Accessibility. The issue with colours being used as the only distinguishing feature in a figure is the problem readers might have with colour-blindness. To counter this one set of histograms in*

Figure 3E could be cross hatched in addition to being a different colour. In Figure 2A the three lines could also be distinguished by being solid, dotted, or dashed in addition to the colour difference.”

Response: we appreciate this helpful suggestion. Following this, we have now modified our figures to make them more accessible to a broad range of readers.

3. *“The green asterisks that mark the Cys mutations in RTX-iv in Figures 4A and 6A should be made larger to stand out more – perhaps twice the diameter.”*

Response: we have made the change following this helpful suggestion.

4. *“The manuscript is well written and clearly described. However, there are few editorial points to make.*

Tris buffers should always stipulate the other buffer component, which is HCl most of the time viz. Tris-HCl.

Page 7, the Figure 3 reference should come a little earlier in the paragraph.

Page 9, three lines down: ‘are resulted’ would be better as – ‘are derived’. Ditto on Page 8, first paragraph.

Page 15, first line of Discussion: critical roles.

Page 16, ‘and so on so forth’ should be: ‘and so on and so forth’. RTX-I should be RTX-i. ‘...disrupting the CR3 binding site or allosterically.’ needs clarification/rewording.

Page 17, 5’ and 3’ should be written with a prime rather than an apostrophe.

Occasionally an article is missing as on page 18, 10 lines down: the C-terminus.

On Page 19, a sentence starts with a numeral (1 μ L), which is to be avoided.

In Acknowledgements the NSERC DG number should be supplied.”

Response: we thank this reviewer for these helpful suggestions. Following them, we have revised manuscript accordingly.

Again, we thank these three reviewers for their constructive and helpful comments and suggestions. We hope that the revised manuscript is now acceptable for publication in *Nature Communications*.

Sincerely yours,

Hongbin Li, Ph. D.
Professor
Department of Chemistry
University of British Columbia
Vancouver, BC V6T 1Z1
Canada
Tel: 604-822-9669
Email: hongbin@chem.ubc.ca

References

- 1 Bumba, L. *et al.* Calcium-Driven Folding of RTX Domain β -Rolls Ratchets
Translocation of RTX Proteins through Type I Secretion Ducts. *Mol. Cell* **62**, 47-62,
(2016).
- 2 Motlova, L., Klimova, N., Fiser, R., Sebo, P. & Bumba, L. Continuous Assembly of β -
Roll Structures Is Implicated in the Type I-Dependent Secretion of Large Repeat-in-
Toxins (RTX) Proteins. *J. Mol. Biol.* **432**, 5696-5710, (2020).
- 3 Carrion-Vazquez, M. *et al.* Mechanical and chemical unfolding of a single protein: a
comparison. *Proc Natl Acad Sci U S A* **96**, 3694-3699, (1999).
- 4 Oberhauser, A. F., Marszalek, P. E., Erickson, H. P. & Fernandez, J. M. The molecular
elasticity of the extracellular matrix protein tenascin. *Nature* **393**, 181-185, (1998).
- 5 Wang, H., Gao, X. & Li, H. Single Molecule Force Spectroscopy Reveals the Mechanical
Design Governing the Efficient Translocation of the Bacterial Toxin Protein RTX. *J. Am.
Chem. Soc.* **141**, 20498-20506, (2019).

REVIEWERS' COMMENTS

Reviewer #1 (Remarks to the Author):

Wang et al. have revised their manuscript, including several key changes that have improved the clarity and comprehension of their work. This manuscript is recommended for publication, provided the following minor revisions are taken into account:

In the Materials and Methods section, on pp. 21-22, the authors write "The MiniTweezers setup is a nonlinear optical trap, and its stiffness varies with the stretching force.⁴⁷ Therefore, F-D curves measured using the MiniTweezers setup cannot be directly converted into force-extension curves of the protein-DNA construct itself." This statement of "nonlinearity" is misleading and of "cannot be directly converted" is incorrect because:

(a) Major nonlinearities in most optical traps are observed when operating in the high force regime (e.g., > 50-60 pN for MiniTweezers, depending on the actual laser power/current setting), where the bead begins to roll off the potential energy well imposed by the laser trap. Therefore, in the regime corresponding to most of the experiments reported in this manuscript, i.e., 0–30 pN, there should be minimal/negligible effect from this common source of nonlinearity.

(b) Below the maximal trapping force and within the harmonic well regime, it is possible—and in fact has been done in many other tweezers studies using MiniTweezers (for example, Bosco et al., NAR, 42:2064-74, 2014 and Huguet et al, PNAS, 107:15431-6, 2010)—to perform trap stiffness calibration so as to convert the pulling curves from Force-Distance (trap position) to Force-Molecular Extension, which can then be directly fitted to the standard WLC model (instead of devising an unconventional and likely misleading pseudo WLC model). More precisely, the stiffness of the optical trap for MiniTweezers can be measured at several different forces within the experimental range, and then fit to the data to resolve any minor force-dependence/non-linearity of the trap stiffness. Using this stiffness calibration curve, one can then directly convert F-D to F-X. Alternatively, following the reference cited by the authors (Ref. 47, see Fig. 17 therein), one can also use the method of video tracking to directly obtain the correct end-to-end molecular extension. Therefore, the authors' claim in the corresponding section and discussion is inaccurate.

However, it is clear that for the purpose of this study, the observable of authors' interest can be obtained by taking the difference in F-D curve at constant forces, where the trap compliance is cancelled out, thereby revealing the actual change in molecular extension from an unfolding or refolding event. Therefore, it is out of simplicity that the authors can proceed with their analyses in the F-D space, where $\Delta D = \Delta x$ is the correct measurement to be considered and fitted to standard WLC. Moreover, the pseudo WLC curves shown in the figures are purely for eye-guiding purpose in helping readers to identify the occurrence of an unfolding or refolding event. It is essential for the authors to be explicit about these details and reasoning instead of stating the rigorous procedure cannot be done.

Several syntactic mistakes/typos need to be fixed. Two of them are mentioned below:

In the second paragraph on p. 3, "After secreted to the extracellular space by the type I secretion system..." could be changed to "After being secreted to the extracellular space by the type I secretion system..." or alternatively "Upon secretion into the extracellular space by the type I secretion system..."

In the third paragraph on p. 3, "Understanding the molecular features underlying the efficient secretion and folding of RTX domain is not only critical for a better understanding of working mechanism of CyaA toxin, but may also help develop new approaches to design therapeutics to combat B. pertussis and other major pathogens." could be changed to "Revealing the molecular features underlying the efficient secretion and folding of RTX domain is not only critical for a better understanding of the working mechanism of CyaA toxin, but may also help develop new approaches to design therapeutics to combat B. pertussis and other major pathogens"

A more detailed caption is needed for Figure S4, such as explaining the observed changes in the CD curves and attributing the sources of signals (i.e., folding states, etc.). Please add n , the number of data points, for each of the 5 waiting time conditions plotted in Figure 5C. On p. 17 in the revised manuscript, the second (ending) citation of Figure S8C in the following sentence seems misplaced and should be taken out. That is, "From the force-dependence of unfolding rate constant, the intrinsic 440 unfolding rate constant k_0 of RTX-iv at zero force in the absence of folded RTX-v was estimated to be 0.11 s^{-1} 441 (Fig. S8C), significantly larger than k_0 of RTX-iv in the presence of folded RTX-v ($1.6 \times 10^{-3} \text{ s}^{-1}$ 442 , Table S1), highlighting the critical role of RTX-v in stabilizing the folded RTX-iv 443 (Fig. S8C)."

Wang *et al.* have revised their manuscript, including several key changes that have improved the clarity and comprehension of their work. This manuscript is recommended for publication, provided the following minor revisions are taken into account:

1. In the **Materials and Methods** section, on pp. 21-22, the authors write “*The MiniTweezers setup is a nonlinear optical trap, and its stiffness varies with the stretching force.*”⁴⁷ Therefore, *F-D* curves measured using the MiniTweezers setup cannot be directly converted into force-extension curves of the protein-DNA construct itself.” This statement of “nonlinearity” is misleading and of “cannot be directly converted” is incorrect because:

(a) Major nonlinearities in most optical traps are observed when operating in the high force regime (e.g., > 50-60 pN for MiniTweezers, depending on the actual laser power/current setting), where the bead begins to roll off the potential energy well imposed by the laser trap. Therefore, in the regime corresponding to most of the experiments reported in this manuscript, i.e., 0–30 pN, there should be minimal/negligible effect from this common source of nonlinearity.

(b) Below the maximal trapping force and within the harmonic well regime, it is possible—and in fact has been done in many other tweezers studies using MiniTweezers (for example, Bosco *et al.*, *NAR*, 42:2064-74, 2014 and Huguet *et al.*, *PNAS*, 107:15431-6, 2010)—to perform trap stiffness calibration so as to convert the pulling curves from Force-Distance (trap position) to Force-Molecular Extension, which can then be directly fitted to the standard WLC model (instead of devising an unconventional and likely misleading pseudo WLC model). More precisely, the stiffness of the optical trap for MiniTweezers can be measured at several different forces within the experimental range, and then fit to the data to resolve any minor force-dependence/non-linearity of the trap stiffness. Using this stiffness calibration curve, one can then directly convert *F-D* to *F-X*. Alternatively, following the reference cited by the authors (Ref. 47, see Fig. 17 therein), one can also use the method of video tracking to directly obtain the correct end-to-end molecular extension. Therefore, the authors’ claim in the corresponding section and discussion is inaccurate.

However, it is clear that for the purpose of this study, the observable of authors’ interest can be obtained by taking the difference in *F-D* curve *at constant forces*, where the trap compliance is cancelled out, thereby revealing the actual change in molecular extension from an unfolding or refolding event. Therefore, it is out of simplicity that the authors can proceed with their analyses in the *F-D* space, where $\Delta D = \Delta x$ is the correct measurement to be considered and fitted to standard WLC. Moreover, the pseudo WLC curves shown in the figures are purely for eye-guiding purpose in helping readers to identify the occurrence of an unfolding or refolding event. It is essential for the authors to be explicit about these details and reasoning instead of stating the rigorous procedure cannot be done.

2. Several syntactic mistakes/typos need to be fixed. Two of them are mentioned below:
 - a. In the second paragraph on p. 3, “*After secreted to the extracellular space by the type I secretion system...*” could be changed to “*After being secreted to the*

extracellular space by the type I secretion system...” or alternatively “Upon secretion into the extracellular space by the type I secretion system...”

- b. In the third paragraph on p. 3, “Understanding the molecular features underlying the efficient secretion and folding of RTX domain is not only critical for a better understanding of working mechanism of CyaA toxin, but may also help develop new approaches to design therapeutics to combat *B. pertussis* and other major pathogens.” could be changed to “Revealing the molecular features underlying the efficient secretion and folding of RTX domain is not only critical for a better understanding of the working mechanism of CyaA toxin, but may also help develop new approaches to design therapeutics to combat *B. pertussis* and other major pathogens”
3. A more detailed caption is needed for Figure S4, such as explaining the observed changes in the CD curves and attributing the sources of signals (i.e., folding states, etc.).
4. Please add n, the number of data points, for each of the 5 waiting time conditions plotted in Figure 5C.
5. On p. 17 in the revised manuscript, the second (ending) citation of Figure S8C in the following sentence seems misplaced and should be taken out. That is, “*From the force-dependence of unfolding rate constant, the intrinsic 440 unfolding rate constant α_0 of RTX-iv at zero force in the absence of folded RTX-v was estimated to be 0.11 s^{-1} 441 (Fig. S8C), significantly larger than α_0 of RTX-iv in the presence of folded RTX-v ($1.6 \times 10^{-3} \text{ s}^{-1}$ 442, Table S1), highlighting the critical role of RTX-v in stabilizing the folded RTX-iv 443 (~~Fig. S8C~~).*”

Reviewer #2 (Remarks to the Author):

The authors addressed most of my concerns even though I regret they did not attempt to examine the unfolding–folding kinetics at more physiological calcium concentrations, as such experiments would have strengthened the present study.

Point-to-Point response to the reviewer's comments is detailed below:

1. *"In the **Materials and Methods** section, on pp. 21-22, the authors write "The MiniTweezers setup is a nonlinear optical trap, and its stiffness varies with the stretching force.⁴⁷ Therefore, F-D curves measured using the MiniTweezers setup cannot be directly converted into force-extension curves of the protein-DNA construct itself." This statement of "nonlinearity" is misleading and of "cannot be directly converted" is incorrect because:*

(a) Major nonlinearities in most optical traps are observed when operating in the high force regime (e.g., > 50-60 pN for MiniTweezers, depending on the actual laser power/current setting), where the bead begins to roll off the potential energy well imposed by the laser trap. Therefore, in the regime corresponding to most of the experiments reported in this manuscript, i.e., 0–30 pN, there should be minimal/negligible effect from this common source of nonlinearity.

(b) Below the maximal trapping force and within the harmonic well regime, it is possible—and in fact has been done in many other tweezers studies using MiniTweezers (for example, Bosco et al., NAR, 42:2064-74, 2014 and Hugué et al, PNAS, 107:15431-6, 2010)—to perform trap stiffness calibration so as to convert the pulling curves from Force-Distance (trap position) to Force-Molecular Extension, which can then be directly fitted to the standard WLC model (instead of devising an unconventional and likely misleading pseudo WLC model). More precisely, the stiffness of the optical trap for MiniTweezers can be measured at several different forces within the experimental range, and then fit to the data to resolve any minor force-dependence/non-linearity of the trap stiffness. Using this stiffness calibration curve, one can then directly convert F-D to F-X. Alternatively, following the reference cited by the authors (Ref. 47, see Fig. 17 therein), one can also use the method of video tracking to directly obtain the correct end-to-end molecular extension. Therefore, the authors' claim in the corresponding section and discussion is inaccurate. However, it is clear that for the purpose of this study, the observable of authors' interest can be obtained by taking the difference in F-D curve at constant forces, where the trap compliance is cancelled out, thereby revealing the actual change in molecular extension from an unfolding or refolding event. Therefore, it is out of simplicity that the authors can proceed with their analyses in the F-D space, where $\Delta D = \Delta x$ is the correct measurement to be considered and fitted to standard WLC. Moreover, the pseudo WLC curves shown in the figures are purely for eye-guiding purpose in helping readers to identify the occurrence of an unfolding or refolding event. It is essential for the authors to be explicit about these details and reasoning instead of stating the rigorous procedure cannot be done."

Response: we appreciate this comment. In principle we agree with this reviewer on these points. As the reviewers correctly pointed out, the stiffness of the optical trap for MiniTweezers can be measured. Based on the measured stiffness, the force-distance curves can be converted to force-extension curves. However, it is worth noting that due to the relatively large size of the beads (~2-3 μm) and the relatively short DNA handle (1.6 kb) used in MiniTweezers experiments, it is known that the trap stiffness may vary with force (even in the range of 20-60 pN) (please see http://tweezerslab.unipr.it/cgi-bin/documents.pl/Show?_id=5eb0&sort=DEFAULT&search=&hits=20). Hence, it is important to measure the stiffness of the trap of MiniTweezers at multiple forces. This can be done using various methods as pointed out by the reviewer. However, in practise, it becomes challenging to do such calibrations for every single tether. Hence, most MiniTweezers studies on protein folding/unfolding have not implemented this approach. Instead, an alternative method is widely used to measure the contour length change ΔL_c upon protein unfolding/folding. We have revised this part in the revised manuscript to made this point clear (see page 18).

2. *“Several syntactic mistakes/typos need to be fixed. Two of them are mentioned below:*
- In the second paragraph on p. 3, “After secreted to the extracellular space by the type I secretion system...” could be changed to “After being secreted to the extracellular space by the type I secretion system...” or alternatively “Upon secretion into the extracellular space by the type I secretion system...”*
 - In the third paragraph on p. 3, “Understanding the molecular features underlying the efficient secretion and folding of RTX domain is not only critical for a better understanding of working mechanism of CyaA toxin, but may also help develop new approaches to design therapeutics to combat B. pertussis and other major pathogens.” could be changed to “Revealing the molecular features underlying the efficient secretion and folding of RTX domain is not only critical for a better understanding of the working mechanism of CyaA toxin, but may also help develop new approaches to design therapeutics to combat B. pertussis and other major pathogens”*

Response: we have corrected these typos/mistakes in the labelling (see page 3 in the revised manuscript).

3. *A more detailed caption is needed for Figure S4, such as explaining the observed changes in the CD curves and attributing the sources of signals (i.e., folding states, etc.).*

Response: following this suggestion, we have included a detailed caption for Figure S4 (see page 3 in the revised supplementary information).

4. *Please add n, the number of data points, for each of the 5 waiting time conditions plotted in Figure 5C.*

Response: we have included the number of data points for all the data points in Fig. 5C (see page 24 in the revised manuscript).

5. *On p. 17 in the revised manuscript, the second (ending) citation of Figure S8C in the following sentence seems misplaced and should be taken out. That is, “From the force-dependence of unfolding rate constant, the intrinsic 440 unfolding rate constant α_0 of RTX-iv at zero force in the absence of folded RTX-v was estimated to be 0.11 s^{-1} 441 (Fig. S8C), significantly larger than α_0 of RTX-iv in the presence of folded RTX-v ($1.6 \times 10^{-3} \text{ s}^{-1}$ 442, Table S1), highlighting the critical role of RTX-v in stabilizing the folded RTX-iv 443 (Fig. S8C).”*

Response: we have made the suggested change (see page 13 in the revised manuscript).

Again, we thank the reviewers for their constructive comments/suggestions, which have helped us improve the clarity of our manuscript.